

# Geodynamic controls on clastic-dominated base metal deposits

Anne Glerum[1], Sascha Brune[1,2], Joseph M. Magnall[1], Philipp Weis[1,2], Sarah A. Gleeson[1,3]

[1]GFZ Potsdam, Potsdam, 14473, Germany
[2]Potsdam Universität, Potsdam, 14469, Germany
[3]Freie Universität, Berlin, 14195, Germany

*Correspondence to*: Anne Glerum (acglerum@gfz-potsdam.de)

**Abstract.** To meet the growing global demand for metal resources, new ore deposit discoveries are required. However, finding new, high-grade deposits, particularly those not exposed at the Earth's surface, is very challenging. Therefore, understanding
the geodynamic controls on the mineralizing processes can help identify new areas for exploration. Here we focus on clastic-dominated Zn-Pb deposits, the largest global resource of zinc and lead, which formed in sedimentary basins of extensional systems. Using numerical modelling of lithospheric extension coupled with surface erosion and sedimentation, we determine the geodynamic conditions required to generate the rare spatiotemporal window where potential metal source rocks, transport pathways and host sequences are present. We show that the largest potential metal endowment can be expected in narrow
asymmetric rifts. This rift type is characterized by rift migration - a process that successively generates hyper-extended crust through sequential faulting, resulting in one wide and one narrow conjugate margin. Rift migration also leads to 1) a sufficient life-span of the migration-side border fault to accommodate a thick submarine package of sediments, including coarse (permeable) continental sediments that can act as source rock; 2) rising asthenosphere beneath the thinned lithosphere/crust resulting in elevated temperatures in these overlying sediments that are favourable to leaching metals from the source rock; 3)
the deposition of organic-rich sediments that form the host rock at shallower burial depths and lower temperatures; and 4) the generation of smaller faults that cut the major basin created by the border fault and provide additional fluid pathways from source to host rock. Wide rifts with rift migration can have similarly favourable configurations, but these occur less frequently and less potential source rock is produced, thereby limiting potential metal endowment. In simulations of narrow symmetric rifts, the potential for ore formation mechanisms is low. Based on these insights, exploration programs should prioritize the
narrow margins formed in asymmetric rift systems; in particular those regions within several tens of kilometres from the paleo-shoreline, where we predict the highest-value deposits to have formed.

**Plain language summary.** Known high-value lead-zinc deposits formed in sedimentary basins created when tectonic plates rifted apart. We use computer simulations of rifting and the associated erosion and deposition of sediments to understand why
they formed in some basins, but not in others. We find that conditions for deposit formation can briefly occur in both narrow and wide rifts for at most 3 My. Our models predict that the largest and the most deposits form in narrow margins of plates that rift asymmetrically.



## 1 Introduction

Decarbonization, population increase, urbanization and a growing middle class mean that current metal reserves will not meet projected future demands (IRP, 2020). Improved recycling will only go partway to meeting this increased demand (e.g., Elshkaki et al., 2018; IEA, 2021) and new ore deposit discoveries will be needed for metals such as copper, zinc, nickel and lead (Lawley et al., 2022; Ali et al., 2017 and references therein). Approximately 50% of global lead and zinc resources are found in sediment-hosted Pb-Zn deposits (Goodfellow et al., 1993; Singer, 1995; Mudd et al., 2017). The highest-value mineral

systems within this broad family of deposits are clastic dominated (CD-type), which are hosted by mixed siliciclastic-carbonate successions in Proterozoic and Paleozoic marine sedimentary basins (Wilkinson, 2014) formed in continental rifts, passive margins and back-arcs (Leach et al., 2005, 2010). Large CD-type deposits can be found in the Mt. Isa Superbasin (Australia), the Kuna and Selwyn Basin (USA and Canada), and other ancient rifted margins in Russia and China (Hoggard et al., 2020). However, there have been very few major CD-type discoveries in the last 25 years (Mudd et al., 2017), and developing

exploration targets for CD-type deposits in ancient sedimentary basins of $10^2$ to $10^5$ km$^2$ in size remains a considerable challenge. This challenge is compounded by the exhaustion of resources located close to Earth's surface (< 100 m; Schodde (2020, 2014)), meaning future discoveries will be located at greater depths. Exploration under several hundreds of meters of cover will require new exploration methods on all spatial scales, from geodynamic to deposit-scale, with an increasing contribution of geophysical, geochemical and conceptional/geological methods (Schodde, 2020).


*Mineral systems* models describe the key processes (often across different scales) involved in the genetic evolution of ore deposits (Hagemann et al., 2016; Mccuaig et al., 2010). In sedimentary basins, key components of the mineral systems model are the metal source, flow conduits for metal-bearing fluids, and a trap for concentrating metals at the deposit site. In CD-type deposits, the primary ore fluid was most likely derived from evaporated seawater, which when heated and circulated at depth

was capable of leaching metals from permeable source (aquifer) units (Leach et al., 2005; Banks et al., 2002) during fluid rock interaction with syn-rift siliciclastic source units, or possibly with basement rocks or mafic volcanics (e.g., Wilkinson, 2023; Ayuso et al., 2004). Compartmentalization of the aquifer units by overlying sag phase sediments could have resulted in locally over-pressured aquifer units that were then periodically breached during periods of extension (e.g., Rodríguez et al., 2021). Syn-extensional faults also provide the main flow conduits for fluids between aquifer units and the site of metal deposition

(e.g., Walsh et al., 2018; Hayward et al., 2021). The deposition of the sulfide mineral CD-type ores is generally concentrated in units undergoing early to burial stages of diagenesis where there is a strong redox gradient, controlled either by a change in stratigraphic unit or by fluid mixing (Wilkinson, 2014). Sedimentary facies that are enriched in organic matter (e.g., mudstones) form particularly effective host rock units in which the required reduced sulfur may be generated via a number of biotic and abiotic processes (e.g., Kelley et al., 2004; Magnall et al., 2018).




Many of the world-class CD-type districts are characterised by long periods of sedimentary basin evolution (> 100 My; e.g. Betts et al., 2006; Beranek, 2017; Gibson and Edwards, 2020), although the underlying geodynamic controls on transient ore-forming events remain unclear. Numerical modelling of hydrothermal fluid flow has focussed on the Mt Isa Superbasin (Yang et al., 2004a, b; Oliver et al., 2006; Sheldon et al., 2019, 2023) and Selwyn Basin (Schardt et al., 2008; Rodríguez et al., 2021).

These basin-scale (< 70 km) hydrothermal studies demonstrate the interplay between the permeability and geometry of the hydrostratigraphic units, heating events and the opening of faults during tectonic events. However, all drivers and facilitators of fluid flow in these simulations greatly rely on assumptions and simplifications of basin, fault and sedimentary layer geometry and other constraints such as thermal architecture. Moreover, deformation of the solid rock medium is often ignored. These conditions are, however, greatly influenced by the dynamics of rifting and the resulting margin architecture and sedimentary

record.

In the field of geodynamics, numerical models have greatly advanced our understanding of sedimentary rift basin formation. The lithospheric strength distribution - dependent on composition, temperature, pressure, strain rate/extension velocity, and weakening mechanisms - controls deformation patterns and the resulting margin architecture (Pérez-Gussinyé et al., 2023;

Brune et al., 2023). For example, coupling of upper crust and mantle lithosphere determines whether deformation occurs over a broad area (for a weak lower crust) or narrow area (strong lower crust; Buck, 1991). Strain-weakening breaks the symmetry of deformation, leading to asymmetric extension (Huismans and Beaumont, 2003). High temperatures and viscous strain softening at the tip of the dominant asymmetric fault and cooling in its footwall generate a strength gradient that sustains rift migration (Brune et al., 2014). New faults continuously form in the direction of migration, accreting material of the opposite

margin to the wide margin left behind. This produces highly asymmetric margins, with the wide margin characterized by hyperextension. Through varying extension velocity, crustal layer thickness, strain weakening and other parameters affecting the lithospheric strength distribution, a range of rifting styles and conjugate margin geometries has been observed in numerical simulations (e.g., Dyksterhuis et al., 2007; Gueydan et al., 2008; Sharples et al., 2015; Tetreault and Buiter, 2018), in particular a metamorphic core complex mode, wide rifting, and narrow rifting leading to symmetric or asymmetric margins.


The more recent coupling of geodynamic modelling software with surface processes codes (Andrés-Martínez et al., 2019; Wu et al., 2019; Beucher and Huismans, 2020; Balázs et al., 2021; Wolf et al., 2021; Neuharth et al., 2022a) has also helped unravel the interaction of sediment (un)loading and rift tectonics. For one, sediment erosion and deposition increases the longevity and offset of faults in the proximal margin, prolongs the different phases of rifting and therefore delays the transition to oceanic

spreading  (Olive et al., 2014; Theunissen and Huismans, 2019; Neuharth et al., 2022a). Sediment loading can also enhance ductile crustal flow in weak crust, promoting sag basin formation (Theunissen and Huismans, 2019). Sediment blanketing maintains higher temperatures in the thinned crust, favouring ductile crustal deformation in the distal margins during late rifting (Andrés-Martínez et al., 2019). Beucher and Huismans (2020) found, however, that for a weak lithosphere, distributed



deformation reduces the accommodation space and the spread-out sediment load will have little feedback on tectonics.

Understanding how different geodynamic and erosional parameters might influence the spatial and temporal alignment of CD-type mineral system components thus has the potential to significantly improve basin prospectivity mapping (e.g., Hoggard et al., 2020; Lawley et al., 2022).

In this contribution, we provide the first numerical modelling study employing a geodynamic model coupled to a surface

processes model to understand ore formation in sedimentary rift basins. Having identified the main components (i.e., source, fluid conduit/fault, trap/host) that characterize the CD-type mineral system, we simulate the spatial and temporal evolution of these components in order to investigate the geodynamic controls on prospectivity. We focus on the role of rift type (narrow vs. wide, asymmetric vs. symmetric), erosion and sedimentation efficiency, and plate velocities in the generation of basins with potential source and host rocks connected by faults. Importantly, we identify three potential ore-deposit forming

mechanisms and find that asymmetric narrow rifting favours all of these mechanisms. We discuss these results in light of known major CD-type deposits as well as their implications for mineral exploration.

## 2 Methods

To investigate the geodynamic controls on CD-type deposit formation, we use a coupled numerical modelling approach that combines the dynamics of continental rifting (Section 2.1) with surface processes (Section 2.2). In Section 2.3 we describe

how the surface processes information is applied to track the sedimentary infill of the tectonically created basins. The sediment types combined with the thermal and mechanical structure of the basins are used to identify ore formation mechanisms in a post-processing step described in Section 2.4. Finally, we list the model assumptions (Section 2.5) and define the model setups (Section 2.6) before presenting the modelling results in Section 3.

### 2.1 Governing equations for continental rifting

Using the ASPECT software (Kronbichler et al., 2012; Heister et al., 2017; Rose et al., 2017; Glerum et al., 2018; Bangerth et al., 2021a, b; Glerum, 2023), we solve for the conservation equations of momentum (1), mass (2) and energy (3) using the extended Boussinesq approximation:

$$-\nabla \cdot [2\eta\dot{\epsilon}(\boldsymbol{u})] + \nabla P = \rho\boldsymbol{g}$$

$$( 1 )$$

$$\nabla \cdot \boldsymbol{u} = 0$$

$$( 2 )$$





$$\overline{\rho}C_P\left(\frac{\partial T}{\partial t} + \boldsymbol{u}\cdot\nabla T\right) - \nabla\cdot k\nabla T = \overline{\rho}H + 2\eta\dot{\epsilon}(\boldsymbol{u}){:}\dot{\epsilon}(\boldsymbol{u}) + \alpha T(\boldsymbol{u}\cdot\nabla P).$$

$$(3)$$

Here $\eta$ is the effective viscosity, $\dot{\epsilon}(\boldsymbol{u})$ the strain rate depending on velocity $\boldsymbol{u}$, $P$ the total pressure, $\boldsymbol{g}$ the gravity vector, $\rho = \rho_0\left(1 - \alpha\left(T - T_{ref}\right)\right)$, $T_{ref}$ the adiabatic reference temperature, $T$ the temperature, $\overline{\rho}$ the adiabatic reference density, $C_P$ the specific isobaric heat capacity, $t$ the time, $k$ the thermal conductivity, $\alpha$ the thermal expansivity, and $H$ the intrinsic specific heat production. $\rho_0$, $C_P$, $k$, $H$ and $\alpha$ are all constant per compositional material.

To track such materials and other fields (e.g., upper crust and plastic strain), we solve an advection equation for each field $c_i$:

$$\frac{\partial c_i}{\partial t} + \boldsymbol{u}\cdot\nabla c_i = q_i(\dot{\epsilon}(\boldsymbol{u})).$$

$$(4)$$

Here the source term $q_i$ is used to update the plastic and viscous strain as $\Delta t\dot{\epsilon}$ in the plastic or viscous domain, respectively, with $\Delta t$ the timestep size.

The effective viscosity in Eq. (1) and (3) is equal to a composite ($\eta_{comp}$) of diffusion (*diff*) and dislocation (*disl*) creep flow

laws when not on yield (Karato and Wu, 1993; van den Berg et al., 1993; Karato, 2008) and equals the plastic viscosity $\eta_p$ based on a Drucker-Prager yield criterion when on yield (Willett, 1992; Davis and Selvadurai, 2002; Glerum et al., 2018):

$$\eta_{diff} = \frac{1}{2}A^{-1}_{w\,diff}d^m\exp\left(\frac{E_{diff} + PV_{diff}}{RT}\right),$$

$$\eta_{disl} = \frac{1}{2}A^{\frac{-1}{n}}_{w\,disl}\epsilon^{\frac{1-n}{n}}_{eff}\exp\left(\frac{E_{disl} + PV_{disl}}{nRT}\right),$$

$$\eta_{comp} = \frac{\eta_{diff}\eta_{disl}}{\eta_{diff} + \eta_{disl}},$$

$$\eta_p = \frac{C\cos(\phi_w) + P\sin(\phi_w)}{2\epsilon_{eff}},$$

$$\phi_w = F_\phi\left(\frac{\max(\epsilon_{min},\min(\epsilon_p,\epsilon_{max})) - \epsilon_{min}}{\epsilon_{max} - \epsilon_{min}}\right)\phi \quad \text{(same for } A_w\text{)}.$$

$$(5)$$

The internal friction angle $\phi_w$ and the viscous prefactor $A_w$ of each material are linearly reduced (weakened) up to a factor $F$ of 0.25 or 0.75 over the plastic and viscous strain intervals of $\epsilon_{min} = 0$ to $\epsilon_{max} = 2$ (e.g., Huismans and Beaumont, 2003; Naliboff and Buiter, 2015; Le Pourhiet et al., 2017). This strain weakening represents frictional softening and viscous strain

softening, respectively, two mechanical weakening mechanisms observed in laboratory experiments and in nature (Bürgmann and Dresen, 2008). The former leads to fault localization, the latter to localization in ductile shear zones, mimicking the drop



in viscosity upon grain-size reduction by large strains (White et al., 1980; Handy, 1989; Rybacki and Dresen, 2004; Austin and Evans, 2009). $E$ is the activation energy, $V$ the activation volume, $R$ the gas constant, $C$ the cohesion, $\epsilon_{eff} = \sqrt{\dot{\epsilon}_{II}}$ the effective strain rate (scalar), with $\dot{\epsilon}_{II}$ the second moment invariant of the deviatoric strain rate tensor. All parameter values can

be found in Table 1.

**Table 1 Material and other model parameters and their symbols, values and units. See Yuan et al. (2019a, b), Guerit et al. (2019), Neuharth et al. (2022a) and references therein for FastScape parameter values.**

| Parameter | Value | | | | | Unit |
|---|---|---|---|---|---|---|
| **ASPECT** | Sediment | Upper crust | Lower crust | Mantle lithosphere | Mantle | |
| Heat capacity $C_P$ | 1200 | 1200 | 1200 | 1200 | 1200 | J kg$^{-1}$ K$^{-1}$ |
| Radioactive heating $H$ | $1.2 \cdot 10^{-6}$ | $1.0 \cdot 10^{-6}$ | $1.0 \cdot 10^{-6}$ | 0. | 0. | W m$^{-3}$ |
| Reference density $\rho_0$ | 2410 | 2613 | 2758 | 3168 | 3182 | kg m$^{-3}$ |
| Thermal conductivity $k$ | 2.1 | 2.5 | 2.5 | 3.0 | 3.0 | W m$^{-1}$ K$^{-1}$ |
| Thermal expansivity $\alpha$ | $3.7 \cdot 10^{-5}$ | $2.7 \cdot 10^{-5}$ | $2.7 \cdot 10^{-5}$ | $3.0 \cdot 10^{-5}$ | $3.0 \cdot 10^{-5}$ | K$^{-1}$ |
| Diffusion prefactor $A_{\text{diff}}$ | $5.97 \cdot 10^{-19}$ | $5.97 \cdot 10^{-19}$ | $2.99 \cdot 10^{-25}$ | $2.25 \cdot 10^{-9}$ | $1.5 \cdot 10^{-9}$ | Pa$^{-1}$ s$^{-1}$ |
| Diffusion activation energy $E_{\text{diff}}$ | 223 | 223 | 159 | 375 | 335 | kJ mol$^{-1}$ |
| Diffusion activation volume $V_{\text{diff}}$ | 0 | 0 | $38 \cdot 10^{-6}$ | $6.0 \cdot 10^{-6}$ | $4.0 \cdot 10^{-6}$ | m$^3$ mol$^{-1}$ |
| Dislocation prefactor $A_{\text{disl}}$ | $8.57 \cdot 10^{-28}$ | $8.57 \cdot 10^{-28}$ | $7.13 \cdot 10^{-18}$ | $6.52 \cdot 10^{-16}$ | $2.12 \cdot 10^{-15}$ | Pa$^{-n}$ s$^{-1}$ |
| Dislocation exponent $n$ | 4 | 4 | 3 | 3.5 | 3.5 | - |
| Dislocation activation energy $E_{\text{disl}}$ | 223 | 223 | 345 | 530 | 480 | kJ mol$^{-1}$ |
| Dislocation activation volume $V_{\text{disl}}$ | 0 | 0 | $38 \cdot 10^{-6}$ | $18 \cdot 10^{-6}$ | $11 \cdot 10^{-6}$ | m$^3$ mol$^{-1}$ |
| Gas constant $R$ | 8.31446 | 8.31446 | 8.31446 | 8.31446 | 8.31446 | J K$^{-1}$ mol$^{-1}$ |
| Cohesion C | $5 \cdot 10^6$ | $5 \cdot 10^6$ | $5 \cdot 10^6$ | $5 \cdot 10^6$ | $5 \cdot 10^6$ | Pa |
| Unweakened friction angle $\phi$ | 26.56 | 26.56 | 26.56 | 26.56 | 26.56 | ° |
| Friction angle weakening factor $F_\phi$ | 0.25 | 0.25 | 0.25 | 0.25 | 0.25 | - |
| Prefactor weakening factor $F_A$ | 0.25 | 0.25 | 0.25 | 0.25 | 1 | - |
| Strain interval $[\epsilon_{min}, \epsilon_{max}]$ | [0, 2] | [0, 2] | [0, 2] | [0, 2] | [0, 2] | - |
| Minimum viscosity $\eta_{min}$ | $5 \cdot 10^{18}$ | $5 \cdot 10^{18}$ | $5 \cdot 10^{18}$ | $5 \cdot 10^{18}$ | $5 \cdot 10^{18}$ | Pa s |
| Maximum viscosity $\eta_{max}$ | $10^{25}$ | $10^{25}$ | $10^{25}$ | $10^{25}$ | $10^{25}$ | Pa s |
| **FastScape** | | | | | | |



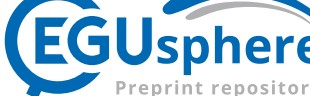

| Marine sedimentation rate $R_M$ | $1\cdot10^{-4}$, $2\cdot10^{-4}$, $4\cdot10^{-4}$ | m yr$^{-1}$ |
|---|---|---|
| River incision rate $K_f$ | $1\cdot10^{-6}$, $1\cdot10^{-5}$, $4\cdot10^{-5}$ | m$^{1-2m}$ yr$^{-1}$ |
| Hillslope diffusion coefficient $K_d$ | $5\cdot10^{-3}$ | m$^2$ yr$^{-1}$ |
| Transport coefficient $K_{M\,sand}$ | 40, 70, 100 | m$^2$ yr$^{-1}$ |
| Transport coefficient $K_{M\,silt}$ | 120, 210, 300 | m$^2$ yr$^{-1}$ |
| Sand and silt porosity | 0 | - |
| Sand e-folding depth | 3700 | m |
| Silt e-folding depth | 1960 | m |
| Depth averaging thickness $L$ | 1000 | m |
| Drainage area exponent $m$ | 0.4 | - |
| Slope exponent $n$ | 1 | - |
| Deposition coefficient $G$ | 1 | - |

## 2.2 Governing equations for surface processes

As described in Neuharth et al. (2022b, a), two-way coupling with the surface-processes code FastScape (Braun and Willett, 2013; Yuan et al., 2019b, a) handles the erosion, transport and deposition of sediment in both the marine and continental parts of the ASPECT model domain (see Figure 1 and Figure 2). These processes are governed by the stream power law with a sediment transport/deposition term, hillslope diffusion and marine transport/deposition. For topography $h$ above sea level (SL) height $h_{SL}$, the following equation is thus used (Yuan et al., 2019a):

$$\frac{\partial h}{\partial t} = U - K_f A^m S^n + \frac{G}{A}\int_A \left(U - \frac{\partial h}{\partial t}\right)dA + K_d \nabla^2 h + \boldsymbol{v}\cdot\nabla h.$$

$$( \, 6 \, )$$

Here $U$ is the tectonic uplift velocity, $K_f$ the incision rate parameter (for the Stream Power Law, SPL), $A$ the drainage area (SPL), $m$ the drainage area exponent (SPL), $S$ the slope (SPL), $n$ the slope exponent (SPL), $G$ the dimensionless deposition/transport coefficient (enriched SPL), $K_d$ the transport coefficient (hillslope diffusion), and $\boldsymbol{v}$ the horizontal velocities used to advect the topography. $U$ and $v$ are supplied by ASPECT at each timestep; all used values for the other parameters can be found in Table 1.

In the marine domain, where $h$ is smaller than $h_{SL}$, the following equation is solved (Yuan et al., 2019b):

$$\frac{\partial h}{\partial t} = K_M \nabla^2 h + \boldsymbol{v}\cdot\nabla h + Q_s + R_M.$$

$$( \, 7 \, )$$



Here $K_M$ is the marine sediment transport coefficient, and $Q_s = \int_A \dot{e}\,dA/(\Delta x \Delta y)$ is the sediment flux coming from the continental domain at points along the shoreline ($Q_s$ is nonzero only along the shoreline), with the erosion rate $\dot{e}$ and the surface area of the whole domain $\Delta x \Delta y$. $R_M$ is the user-set background marine sedimentation rate that deposits a homogenously thick layer of sediments in the marine domain.

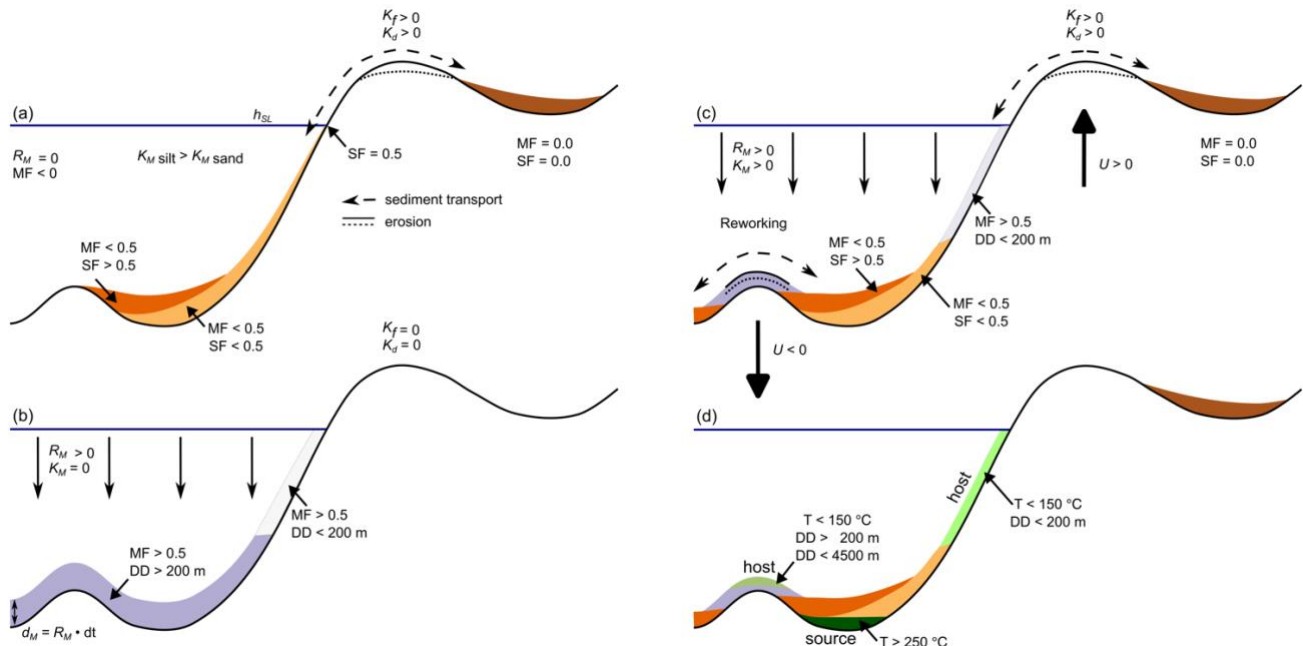

*Figure 1 Schematic illustrations of the sedimentary processes in our coupled ASPECT-FastScape workflow and the parameters used in Eq. (6)-(8). (a) Endmember case of purely continental sediment erosion, transport and deposition in the continental and marine domain. (b) Endmember case of purely marine sedimentation. (c) Combined continental and marine erosion, transport and deposition, including reworking, as used in this work. (d) Definitions of continental source and marine host rock units.*

**2.3 Formation of basin stratigraphy**

The ASPECT side of the coupling to FastScape tracks the deposited sediment and its age and deposition depth (DD) on compositional fields that are advected according to Eq. (4) (Neuharth et al., 2022a). We have extended the interface to also track 1) the dominant grain size of the continental sediment (coarse or fine, using FastScape's 'silt fraction', hereafter called SF) within the marine domain and 2) the fraction of marine sediments (MF) of the sum of marine and continental sediments deposited within one ASPECT timestep at a particular location along the submarine surface. On land, FastScape's silt fraction SF is 0 everywhere; upon entering the marine domain sediments receive the default value of 0.5 (Figure 1a). Due to the different transport properties of silt and sand within the marine domain ($K_M$ and the 'e-folding depth', see Table 1), this fraction



then evolves with transport distance from the shoreline (staying between 0 and 1). Marine sediments are deposited everywhere in the sub-sea-level domain at a user-specified rate $R_M$ by increasing the ASPECT surface height in the marine domain before passing it to FastScape (Figure 1b). The increase is equal to the timestep size times the deposition rate, but the new height cannot exceed the sea-level height. The fraction of marine sediments MF out of the total thickness of sediments deposited in one location in one ASPECT timestep is computed as

$$MF = \frac{h_{marine} - h_0}{h_{total} - h_0 - U \cdot \Delta t}.$$

$(\ 8\ )$

Here $h_0$ is the height above the initial ASPECT surface at the beginning of the ASPECT timestep, $h_{marine}$ is the height after marine sediments have been deposited, $h_{total}$ the height that has been subsequently returned by FastScape (including uplift, erosion and sedimentation effects), $U$ the uplift velocity provided to FastScape and $\Delta t$ the ASPECT timestep size. If material has been eroded (i.e., $h_{total} - U \cdot \Delta t - h_0 < 0$), $MF$ is set to zero.

### 2.4 Proxies for mineral system components and metal endowment

The general model for CD-type metallogenesis describes the leaching of metals from a source rock by fluids, transport of these fluids upwards along faults, and deposition of the metals they carry in units undergoing early to burial stages of diagenesis in marine sedimentary basins (e.g., Wilkinson, 2003; Heinrich and Candela, 2014; Magnall et al., 2021). As base metal solubilities increase with temperature and salinity (Yardley, 2005), evaporated seawater that has been heated up has the potential to leach and transport high concentrations of metals (e.g., Banks et al., 2002; Wilkinson et al., 2009). Leaching occurs at depth through fluid-rock interaction with the intra-basinal syn-rift siliciclastic rocks such as sandstones (Ayuso et al., 2004; Leach et al., 2005). The bottom sedimentary sequences and the fluids flowing through them can be heated up by temperature anomalies generated by rifting (e.g., through proximity to hot asthenosphere after thinning of the lithosphere), driving thermal fluid convection allowing for ore-forming fluid temperatures between 150 and 300 °C (e.g., Cooke et al., 2000; Magnall et al., 2016), which can result in Zn and Pb solubilities of 100s to 1000s ppm (e.g., Wilkinson et al., 2009). Fault activity (e.g., seismic pumping and pulsing and fault valving) and passive upward flow then bring the fluids to the surface (Walsh et al., 2018). Host rocks to the mineralization are typically carbonaceous marine mudstones and siltstones with some carbonate units (e.g., Ridley, 2013; Wilkinson, 2014). In this syn-diagenetic environment, base metal solubilities decrease along steep thermal and chemical gradients. When the fluids encounter reduced sulfur, sulfide minerals are formed, which are only stable under reducing conditions (Cooke et al., 2000; Ridley, 2013). Sedimentary facies that are enriched in organic matter (e.g., mudstones) therefore form particularly effective host rock units as they can produce large volumes of reduced sulfur via a number of biotic and abiotic pathways (e.g., Kelley et al., 2004; Magnall et al., 2018). For bacteriogenic (biotic) processes to occur, maximum temperatures of the host rock should be below ~110 °C (Takai et al., 2008; Ridley, 2013). At temperatures over 110 °C, abiotic pathways of sulfate reduction prevail, e.g., thermochemical sulfate reduction (Machel, 2001).



We translate the above description of the CD-type mineral system into a series of simplified conditions that identify the main components of the system in our simulations, using the predicted dominant sediment type, its deposition depth and the temperature field. These conditions are evaluated in a postprocessing step, in which the ASPECT results are analysed using ParaView (Ahrens et al., 2005; Ayachit, 2015) python scripts as follows:

1. Based on the tracked fields described in Section 2.3, we first separate sediments from basement materials (e.g., upper and lower crust). We then further divide and colour code the sediments into five types: 1) sandstone (where silt fraction SF is smaller than 0.5 and marine fraction MF is smaller than 0.5; yellow colours in the figures), 2) siltstone (where SF >= 0.5 and MF < 0.5; orange), 3) limestone (where MF >= 0.5 and deposition depth DD below sea level < 200 m; light purple), 4) carbonates (where MF >= 0.5 and 200 m <= DD < 4500 m; medium purple) and 5) silicious marine sediments (MF >= 0.5 and DD >= 4500 m, which represents the carbonate compensation depth (e.g., Middelburg, 2019); dark purple). Note that sediment types are classified according to their dominant component (Figure 1c). For example, when sediment is labelled as sandstone, it means that the dominant (i.e., more than 50%) grain size is coarse and from a continental source region, not that no continental fine-grained or marine sediment is present. When sediments in the marine domain are eroded after being deposited, their previous type is lost and they are classified as continental in the computation of MF even if they were labelled as marine before.

2. Combining these sediment types and the temperature field, we identify and colour code potential source and host rock (Figure 1d). Where temperatures in the sandstone exceed 250 °C, it is coloured dark green, indicating source rock conditions favourable for enhanced metal leaching and the generation of ore-forming solutions. Where marine sediments (types 3-5) are below a temperature of 150 °C, they are coloured light-green, indicating potential host rock, i.e., where redox becomes the predominant factor controlling metal solubilities.

3. Potential deposit-forming mechanisms are then defined as those events where an active fault (high strain rate; approximately $5 \cdot 10^{-15}$ 1/s and above, see grey-scale in figures) or an inactive fault (high plastic strain of ~3 and above, low strain rate) connects source and host rock within the same or neighbouring basins. Ore-forming fluids will precipitate metals when they ascend along a fault and first encounter prospective host rock units (the thickness of the host rock is thus less important than the area of source rock). We assume that active faults are always more permeable than the surrounding rock and therefore act as preferred upward pathways for fluids. However, as summarized by Walsh et al. (2018), faults also facilitate replenishing downward flow of fluids, and their permeability can be greatly decreased by for example the generation of clay smears. Shearing of unlithified sediments with high clay content (like CD-type host rock) can lead to lower permeabilities than in the surrounding shallow (< 3 km) sediments (e.g., Barnicoat et al., 2009; Neuzil, 2019; Sheldon et al., 2023). For a discussion of fault permeability, see also Bense et al. (2013) and Gleeson and Ingebritsen (2016).





Within ASPECT we also compute at each timestep the total area of potential source and host rock, using the same definitions as above and integrating over each mesh element. Based on the total source rock area $A$, we estimate the possible metal endowment $E$ as:

$$E = A \cdot dz \cdot \rho \cdot F_{leaching} \cdot C_{initial},$$

*( 9 )*

where $\rho$ is the rock density, $dz = 20$ km is the assumed in-plane extent of the basins to create a source rock volume, $F_{leaching}$ is the leaching capacity factor assumed to be 0.65 (from Windermere Supergroup leaching experiments; Lydon (2015)) and $C_{initial}$ is the initial metal concentration of the source rock, taken to be 100 ppm for Zn and 23 ppm for Pb (Lydon, 2015).

**2.5 Model assumptions**

The coupled geodynamic-surface processes model cannot explicitly resolve hydrothermal fluid flow and fluid-rock
interactions, therefore the proxies for metal endowment provide maximum estimates under the simplifying assumptions that 1) the source rocks are permeable enough to support metal transport by fluid flow, 2) the hydrology provides an efficient focusing mechanism along fault structures, 3) the thermal structure remains close to the conduction-dominated profile calculated by the geodynamic model during most of the ore-forming event, and 4) the availability of reduced sulfur from the host rock is sufficient to provide an active metal trap.


Although our workflow does not directly account for the above sub-fault-scale processes, it nevertheless captures the essence of CD-type deposit formation by identifying source and host rocks and fluid migration pathways. As such, this workflow uses the same solution to the cross-scale problem as petroleum and mineral system analysis (Allen and Allen, 2013, Chapter 11; Mccuaig et al., 2010). Similar workflows to deduce the first-order impact of fluid flow and associated geochemical alteration
from numerical forward models have shown to be successful avenues for estimating hydrogen generation at mid-ocean ridges (Rüpke and Hasenclever, 2017) and rifted margins (Liu et al., 2023), and carbon dioxide release at extensional plate boundaries (Hasenclever et al., 2017; Gorczyk and Gonzalez, 2019). To truly cross the scales from large-scale dynamics to mineralization, the inclusion of hydrothermal flow and reactive-transport modelling (e.g., Yapparova et al., 2019) into the combined modelling of geodynamic and surface processes as presented in this paper is required.


While our simulations account for self-consistent interactions between thermal, mechanical, and sedimentary processes, they also involve several geodynamic limitations. For one, they are 2D and therefore ignore potential along-strike effects on the structural heterogeneity of the rift and margins (e.g., Naliboff et al., 2020; Gouiza and Naliboff, 2021), effects of oblique extension (Brune et al., 2012, 2018) and, on a smaller scale, they ignore the possible compartmentalization of subbasins into
a complex, multi-generational, multi-directional 3D fault geometry as observed in the McArthur Basin (Australia) that hosts several world-class deposits (Blaikie and Kunzmann, 2020). Furthermore, as the initial, Proterozoic state of the lithosphere is



mostly unknown, we start off our simulations with a relatively smooth initial lithosphere geometry and approximate all possible sources of tectonic inheritance with initial plastic strain (see Section 2.6). Finally, we do not include melting and magma flow, and thus also ignore their effect on heat flow, faulting, the overall stress state, the generation of oceanic crust and its potential
role as metal source rock (e.g., Murphy et al., 2011).

### 2.6 Model setup

The model setup represents a 2D extensional lithosphere-asthenosphere system (for ASPECT input files and source code, see Glerum (2023)). The model domain spans 700 x 300 km and includes - in the reference case - a 20-km thick upper crust, a 15-km thick lower crust and 85 km of mantle lithosphere, see Figure 2. The rheologies assigned to these materials are wet quartzite
(Rutter and Brodie, 2004), wet anorthite (Rybacki et al., 2006), dry olivine (Hirth and Kohlstedt, 2003), and wet olivine for the asthenosphere (Hirth and Kohlstedt, 2003), respectively. See Table 1 for the properties of each material layer. The upper crust and the mantle lithosphere are locally thickened or thinnned around x = 350 km using a Gaussian shape with a standard deviation of 60 km. To further localize deformation, the plastic strain field is initialized with random noise with a maximum amplitude of 0.25 centred at x = 350 km (see Figure 2a).


The initial temperature follows a 1D continental steady-state geotherm (Chapman, 1986) at each x-coordinate, constrained by a Lithosphere-Asthenosphere Boundary (LAB) temperature (coinciding with the bottom of the mantle lithosphere field at the start of the model run) of 1350 °C and a surface temperature of 20 °C (see Table 1 for other relevant parameters). Below the lithosphere, the initial temperature follows an adiabat with a surface potential temperature of 1303 °C, such that at the LAB
the geotherm and adiabat match. As such, we assume that the tectonic settings of the Paleoproterozoic in which most CD-type deposits formed can be represented by a setup representative of present-day continental rifting.

Extension is prescribed through outward horizontal velocities on the left and right vertical boundaries (see Figure 2a). The reference total prescribed horizontal velocity is 10 mm yr$^{-1}$, which resembles the velocity of the initial slow rift phase (Brune
et al., 2016) and falls in the ultra-slow range that produces magma-poor margins (Pérez-Gussinyé et al., 2023). On the bottom boundary, the initial lithostatic pressure below reference lithosphere at 300 km depth (at x = 1 km) is prescribed as the normal traction component, such that material is free to enter or leave the domain in response to the internal stress state. Temperature and composition are fixed to their initial values at this boundary (i.e., the inflowing material is asthenosphere). The surface boundary is deformed using FastScape (see Section 2.2), with full stabilization according to Kaus et al. (2010) and Rose et al.
(2017). The FastScape sea level is set to 200 m below the initial surface of the model domain, so at $y = 299.8$ km, where $y$ is the vertical upward coordinate. Along the deforming surface boundary, the fixed boundary condition for the deposition depth (DD) field is set to $DD = 299.8 - y$ (in km), the sediment age field to time in My and the sediment field is set to one, other fields are kept fixed at a value of zero. Along the bottom boundary, all fields are fixed at zero.



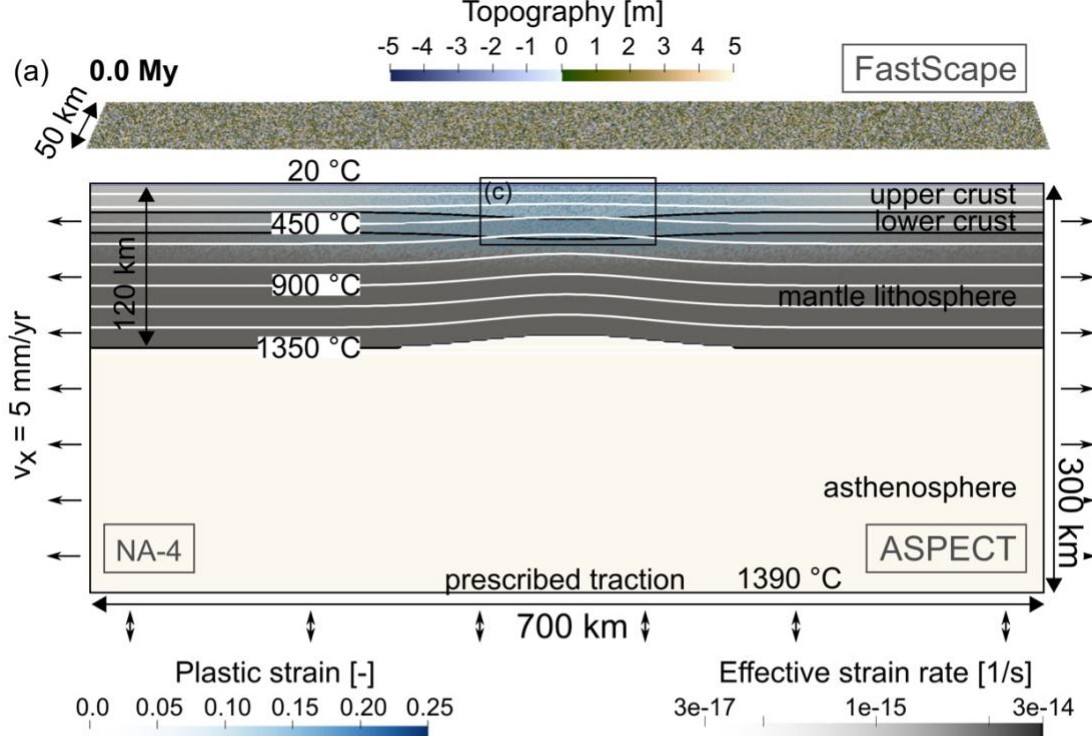

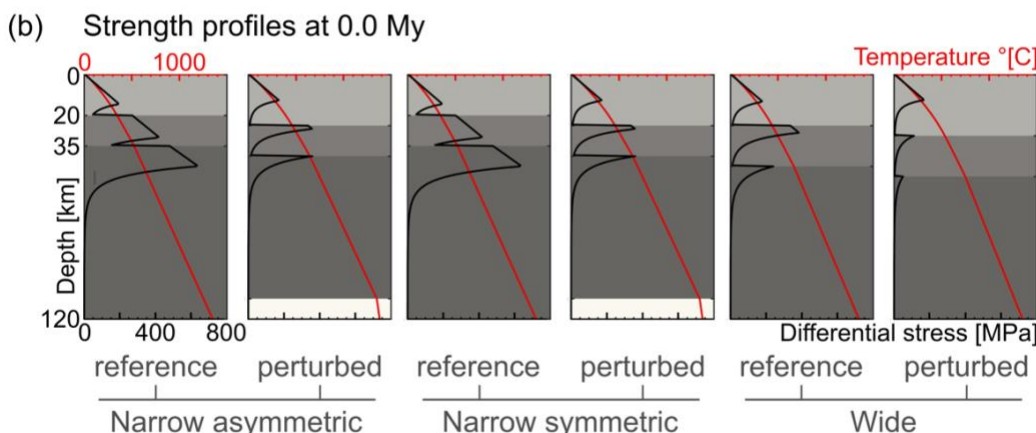

(b) Strength profiles at 0.0 My

(c) Strain rate fault patterns at 1.5 My

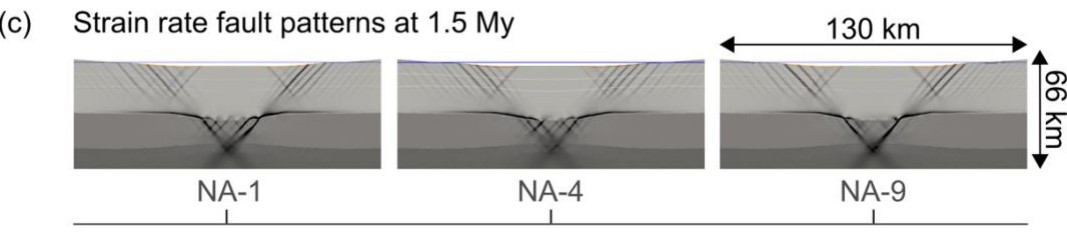



*Figure 2 (Previous page) Numerical model setup. (a) ASPECT and FastScape model domains for reference model NA-2, showing initial composition (beige and grey areas), initial temperature (white isotherms), initial plastic strain (blues), boundary conditions and initial FastScape topography (blues and greens) with respect to sea level at -200 m from the initial ASPECT model surface at y = 300 km. Rifting driven by the prescribed tensional horizontal velocities is localized initially in the centre of the domain due to the higher amplitude plastic strain and thicker upper crust. The bottom boundary of the ASPECT domain is open to flow, its normal traction is prescribed the initial lithostatic pressure at x = 10 km, y = 0 km. FastScape is initialized with a random topography of ±5 m. (b) Temperature and strength profiles in unperturbed and perturbed (centre of domain) locations for each of the three modelled rift types: narrow asymmetric (NA), narrow symmetric and wide rifting assuming an effective strain rate of $10^{-15}$ s$^{-1}$. (c) Strain rate patterns in the crust at the horizontal centre of the domain after 1.5 My for three out of the nine different initial plastic strain configurations of the narrow asymmetric rift type. The effective strain rate is defined as $\sqrt{(|(|\dot{\epsilon}|_{II})|)}$, the square root of the second moment invariant of the deviatoric strain rate.*


The simulations presented in the following section differ in the strain weakening factor $F$ (see Eq. (5)), the thickness of the lithospheric layers, the initial perturbation of these layers to focus deformation, surface process parameters, and the initial plastic strain. By varying the first three parameters, we obtain three end-member rifting styles: narrow asymmetric, narrow symmetric, and wide rifting (e.g., Huismans and Beaumont, 2003; Brune et al., 2014; Svartman Dias et al., 2015). For the

narrow asymmetric cases, the upper crust is locally thickened to 25 km and the mantle lithosphere locally thinned by 15 km at the domain's centre (see Figure 2b). For narrow symmetric rifting, strain weakening is reduced by increasing the weakening factor $F$ from 0.25 to 0.75 with respect to the narrow asymmetric cases. Wide rifting is obtained for an upper crust of 25 km, a lower crust of 20 km and a mantle lithosphere layer of 75 km of which the upper crust is locally thickened to 30 km and the mantle lithosphere thinned to 70 km. Weakening factor $F$ is again 0.25. For the narrow asymmetric simulations, we

subsequently vary the surface process parameters $R_M$, $K_M$ and $K_f$ and the extension velocity.

For each of the above cases, nine simulations are run with different initial plastic strain configurations. The initial strain configuration determines the early fault pattern and coalescence, as shown for three simulations in Figure 2c, and therefore the subsequent co-evolution of thinning and faulting. The initial strain represents possible tectonic inheritance, such as foliation,

LPO-induced mechanical anisotropy (Tommasi and Vauchez, 2001) and other pre-existing structures where continental rifts prefer to localize. The nine initial configurations account for possible along-strike variability in inhered heterogeneity (Richter et al., 2021) and allow us to assess the robustness of the modelling results. To this end, we both discuss results of individual simulations and report averages of, for example, the number of basins favourable to ore formation and area of source rock.

Each simulation runs for 25 My of model time, which is enough to reach continental break-up, after which the syn-rift, syngenetic and syn-diagenetic CD-type deposits are no longer expected to form. Although studies have suggested a post-rift





or syn-inversion timing for some CD-type deposits (e.g., Gibson and Edwards, 2020), syn-rift deposits have been most comprehensively studied and are the focus of our study.

## 3 Results

We first present the overall rift evolution and describe the favourable conditions for ore formation in the simulated sedimentary basins for each rifting type (narrow asymmetric, narrow symmetric and wide rifting). Then we investigate the effect of varying surface process parameters and extension velocity on source and host rock creation in narrow asymmetric rifts.

### 3.1 Narrow asymmetric rifts

Out of the nine simulations of narrow asymmetric rifting (simulations NA-1 to NA-9), the representative temporal evolution
of simulation NA-4 is presented in Figure 3. After an initial phase of distributed deformation (Figure 3a&g), by 2.5 My two conjugate normal faults have formed. By 5 My (Figure 3b&h), the currently active left border fault (black colours indicating high strain rates) extends to the base of the lithosphere and new faults are forming in the rift centre. Rift shoulder topography (maximally 2139 m) has reached its highest point and sedimentary basins are being filled with mostly coarse continental sediments (sandstone; yellow areas). At 10 My, the rift centre has started migrating towards the right, and the old left shear
zone is abandoned (Figure 3c&i; blue colours indicating high plastic strain). An exhumation channel runs from the uprising asthenosphere to the active faults, along which lower crust flows up from the narrow margin. This counterbalances the thinning of the crust by the new faults, postponing continental breakup (e.g., Brune et al., 2014). Marine sediments (shallow limestones) deposited in the rightmost basin (Figure 3i; shown and labelled as light-green potential host rock) are subsequently buried by new continental sediments (Figure 3j). Rift shoulder topography is eroded and the sequential faulting of rift migration
continues. By 15 My, the rightmost basin has heated up enough for the lowermost sandstone layers to be classified as source rock (dark green area in Figure 3j). They are connected by the now inactive border fault to the marine sediments higher up in the stratigraphy, which are cool enough to act as host rock (light green). This constellation aligns potential source rock, fluid pathway and host rock and could promote ore formation by hydrothermal fluid flow. As the border fault is inactive, the constellation is not the most favourable constellation encountered in the NA simulations and we therefore term these conditions
'ore formation mechanism 3', see Figure 4c.







*Figure 3 (Previous page) Model evolution of narrow asymmetric rift simulation NA-4. (a)-(f) Time series of whole domain showing different materials, isotherms (white), strain rate (grey scale), sediment type (orange, yellow and purples), and velocity (brown vectors). Black rectangles indicate zoomed-in areas in (g)-(l). (g)-(l) Time series of zoom-ins focussing on FastScape topography (top) and sedimentary basins (bottom), additionally showing plastic strain (blue scale) and sediment age (brown contours in steps of 5 My). Coarse continental sediments (sandstone) that exhibit temperatures higher than 250 ℃ are classified as source rock and coloured dark green. Marine sediments at temperatures below 150 ℃ are potential host rocks for metal deposition and are coloured in light greens. The black rectangle in (k) indicates the location of a further zoom-in in Figure 4.*

Five million years later (Figure 3k), the border fault has been reactivated, but the marine sediments along the fault are no longer in the host temperature window (represented by the colour change from light green to light purple in Figures 3j&k). However, the sequential faulting associated with rift migration now leads to smaller synthetic faults fracturing the major basin adjacent to the border fault, tapping the potential source rock, which has grown in size during basin evolution, and providing a pathway to the potential host rock lying above. We consider these conditions more favourable, because a larger source rock region is now connected to an overlying host rock by active faults instead of an inactive fault, so we label them as 'ore formation mechanism 1' (Figure 4a). At the same time, one of the deeper basins on the wide margin develops and maintains a small pocket of source rock (Figure 3e-k), because the rising asthenosphere that has replaced the thinning lithosphere maintains higher temperatures in the extended crust above. This source rock is connected to overlying host rock by an inactive fault; this constitutes another instance of ore formation mechanism 3. Rift migration then ceases after 20 My and the right border fault becomes inactive. By 22 My, oceanic spreading has started and highly asymmetric margins have formed (Figure 3f&l). Source rock can still be found at the sea floor of the rightmost basin, in the spreading centre and in one basin on the wider left margin, while host rock covers large parts of the wider margin. However, at this stage, no more potential ore-forming mechanisms can be identified.

The nine narrow asymmetric rifting simulations with varying initial plastic strain configurations evolve very similarly. All simulations produce asymmetric margins, with five simulations showing rift migration towards the left and four towards the right (i.e., the direction of migration is random). The amount of sediments deposited increases approximately linear over time and totals on average 962 km$^2$ after 25 My. Only a fraction of those sediments actually could serve as source ($\leq$ 21.9 km$^2$; Figure 5a) or host ($\leq$ 271.7 km$^2$) rock, either because it is not the right type of sediment or not in the right temperature window. Whereas the amount of host rock increases steadily over time, source rock only starts to form after 8 My (in or close to the rift centre, Figure 3i and Figure 5a) when the rift shoulders have provided significant sediment supply to the marine domain. Source rock area massively increases from around 15 My to its maximum around 20 My (for seven out of the nine simulations) as source rock forms in the major basin on the narrow margin, before slowly declining during the last ≤5 My (with the exception of NA-7, Figure 5a). The decline is temporally related to the stabilization of the rift centre (starting between 17.5



and 21.5 My except for NA-7) and its transition to oceanic spreading (starting from 20 My, grey area in Figure 5a, except for NA-7).

*Figure 4 The three ore formation mechanisms observed in simulations of narrow asymmetric rifting, i.e., scenarios favourably combining source and host rock and fault activity as required for metallogenesis. The upper- and lowermost rows show all rift basins, the zoom-ins in between only the relevant subbasin over time. Note that the zoom-in location changes over time to stay centred on the basin. White arrows and annotations indicate the predicted depth of mineralization at the bottom of host rock sequences. Blue arrows and annotations indicate the minimum distance of the mineralization to the shore line.*



The model results contain three resolvable parameters (c.f. Section 2.4) that are critical for the overall mineral system: 1)
source rock for fluids to leach metals from; 2) host rocks for fluids to deposit the metals upon reaching them; and 3) active
and/or inactive faults that act as fluid pathways connecting source and host rock. Figure 4 zooms in on the three ore formation
mechanisms combining these ingredients that occur in the suite of nine narrow asymmetric rift simulations. Ore formation
mechanism 1 (Figure 4a, also Figure 3k) shows a scenario in which the faults of the rift centre migrate into the major active
basin on the narrow margin. This basin contains both source and host rock, which are subsequently connected by the new
faults providing upward fluid pathways. Ore formation mechanism 2 (Figure 4b) shows how the major basin on the left narrow
margin develops both source and host rock that are connected by the reactivated (or continuously active) border fault. In ore
formation mechanism 3 (Figure 4c), the source rock forming in a basin is connected to host rock by an inactive fault. This can
occur both in the wider margin, and in the major basin (Figure 3k and j, resp.).

To summarize the occurrence of components required for ore formation, we count the number of basins in which (1) source
rock occurs, (2) source and host rock co-occur, and in which (3) source and host rock are connected by an inactive (ore
formation mechanism 3) or (4) an active fault (mechanisms 1 and 2). This analysis is performed at 2.5 My intervals. The
maximum counts are averaged over the nine simulations with different initial strain configurations and plotted in Figure 6. On
average, narrow asymmetric rifts contain two basins that align with the most prolific ore formation mechanisms 1 and 2, i.e.,
they contain source and host rock connected by active faults.

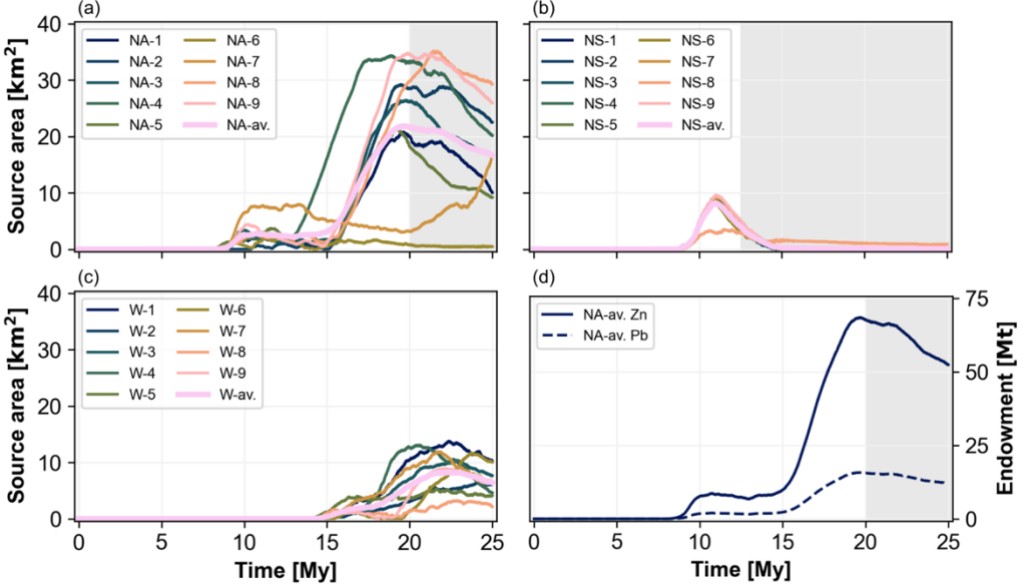

*Figure 5 Source area over time for each simulation plus their arithmetic average for each rift type. Grey areas indicate the period of time when oceanic spreading occurs in the individual simulations. (d) The narrow asymmetric (NA) average is used to compute zinc and lead endowment over time.*



## 3.2 Narrow symmetric rifts

A representative narrow symmetric simulation is shown in Figure 7 (NS-8 out of NS-1 to NS-9). Initially, the symmetric rift evolution resembles the asymmetric one. However, the phase of distributed faulting is prolonged compared to narrow

asymmetric rifting (compare Figure 7b and Figure 3b) and a symmetric sedimentary infill of sand, silt and shallow marine sediments is deposited in the area of deformation. Similarly high rift shoulders develop, with a maximum topography of 2070 m. By 10 My, two major faults dipping away from the rift centre accommodate most of the deformation and the lithospheric mantle has broken up. Along both large shear zones, a small pocket of source rock forms that is connected to host rock deposited at great water depth through antithetic faults (ore formation mechanism 1). By 15 My, continental

breakup has occurred and oceanic spreading starts. Marine sediments continue to be deposited away from the spreading centre, and a small amount of coarse continental sediments in the oceanic spreading centre is hot enough to be labelled as source rock (Figure 7d). However, no favourable conditions for ore formation occur after breakup. On average, less than one mechanism 1 or 2 basin occurs in the narrow symmetric rift evolutions, and the average maximum amount of source rock is only about one-third of the narrow asymmetric source rock (Figure 6). Source rock area soon returns to zero after oceanic

spreading has started (Figure 5).

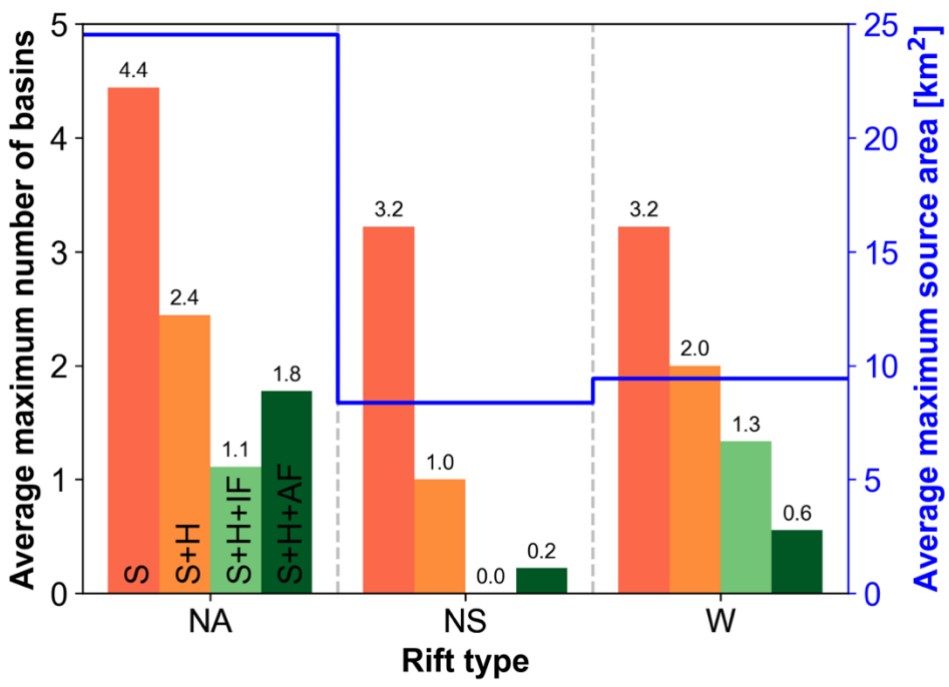

*Figure 6 The average number of basins that contain source rock (red, labelled "S"), source and host rock (orange, S+H), and source and host rock connected by inactive faults (light green, S+H+IF) or active faults (dark green, S+H+AF) per rift type (NA – narrow asymmetric, NS – narrow symmetric, W – wide). The blue line plots the average maximum source area per rift type, which eventually determines the size of the potential mineral resource. The narrow asymmetric rift type produces the most source rock and has the most basins with active faults that provide fluid pathways between source and host rock. All averages are taken over the nine initial strain configurations per rift type.*





### 3.3 Wide rifts

A typical simulation (W-2) of the initially wide rifting simulations (W-1 to W-9) is presented in Figure 8. A broader area of deformation leads to lower rift shoulder topography (maximally 868 m) and a shallower, but broader marine domain than in the previous simulations. By 10 My, 5 larger faults shoal into the upper-lower crustal shear zone. The basins are predominantly

filled with sand and shallow marine sediments. Five million years later, the mantle lithosphere has almost thinned to breakup (Figure 8d) and the development of source rock starts (Figure 5c). Soon after, the rift migrates towards the left (Figure 8e). By this time, two major basins on the deserted margin have heated enough to generate source rock connected to host rock by inactive faults (Figure 8i). This ore formation mechanism 3 persists at least until 25 My, when the rift centre has stabilized again (Figure 8f&j). The asymmetric margins also exhibit asymmetry in terms of sediment type, with the narrower left margin

hosting more fine-grained continental sediments and the wider margin hosting extensive younger marine sediments. The average maximum amount of source rock (~9.4 km$^2$) is again smaller than in the narrow asymmetric rift (Figure 5&6). Additionally, Figure 6 shows that fewer active faults connect source and host rock than for narrow asymmetric rifts.

### 3.5 Varying sedimentation and erosion efficiency

Surface process efficiencies can vary significantly as a function of lithologic, environmental and climatic factors and are often

not well constrained. By changing key parameters, we therefore assess their relative impact on model results. Starting from the model setup for narrow asymmetric rifting, which leads to the most prolific ore formation mechanisms (as shown by Figure 6), we first vary the pelagic sedimentation rate, then the efficiency of marine and of continental erosion and finally the extension velocity. We show changes in model results by comparing the total source and host area over time averaged over the nine simulations with different initial plastic strain configurations (Figure 9).

### 3.5.1 Varying marine sedimentation rate

Both source and host rock total area increase when the background marine sedimentation rate is increased (Figure 9a). Source rock starts forming at around 10 My in each case, but the period of fastest increase starts earlier and lasts longer the higher the marine sedimentation rate. The approximately linear increase in host rock has a larger slope for higher rates, as would be expected. The higher source rock area for a sedimentation rate of $4 \cdot 10^{-4}$ m yr$^{-1}$ is reflected in the occurrence of source rock

at in every basin at 20 My (Figure 9a top), not just in the basin in the narrow margin. However, these basins only experience ore formation mechanism 3; no active faulting occurs in the wide margin. Ore formation mechanisms 1 and 2 still occur in the major basin on the narrow margin. The overall potential to form deposits has therefore increased.



*Figure 7 Model evolution of narrow symmetric rift simulation NS-8 showing continental breakup after 15 My and the development of equally wide margins. (a)-(e) Time series of zoom-ins focussing on FastScape topography (top) and sedimentary basins (bottom). The figures are always horizontally centred around x = 350 km. Black rectangles indicate the locations of further zoom-ins in (f)-(h). (f)-(h) Zoom-ins of specific basins with and without ore formation mechanisms. (g) Ore formation mechanism 1, where new active faults connect source and host rock in neighbouring basins.*



### 3.5.2 Varying marine erosion efficiency (diffusivity)

Increasing the marine diffusion coefficient has a very limited effect on the amount of host rock, but it leads to an increased generation of source rock in the major basin on the narrow margin (Figure 9b). A larger diffusivity leads to a faster distribution of sediments away from the continent and therefore to a larger distal sediment load. Source rock area therefore occurs more often in the wide margin (see example in Figure 9b top). This means that ore formation mechanism 3 could potentially produce larger CD deposits than predicted in the reference narrow asymmetric simulations.

### 3.5.3 Varying continental erosion efficiency

An increased continental erosion efficiency (Figure 9c) translates to a larger supply of continental sediments to the marine domain. As expected, this is mirrored in a larger amount of total sediments and a larger amount of source rock. There is also an increase in host rock, but this is relatively smaller. The high sediment supply of very efficient erosion ($K_f = 4 \cdot 10^{-5}$ m$^{1-2m}$ yr$^{-1}$) seems to prolong rift migration: where for the other erosion efficiencies the rift stabilizes in seven and eight of the nine simulations, respectively, in the high $K_f$ case, only four do within 25 My. In the high $K_f$ case, continental sediment supply during the erosional phase that removes rift shoulder topography is so large that close to the shore no marine sediments are deposited (the accommodation space up to sea level is completely filled by continental sediments). Therefore, the ore formation mechanism 2 phase is shortened or absent in all simulations (Figure 9c bottom). Mechanism 1, on the other hand, occurs in all five prolonged rift migration simulations around 22.5-25 My.

### 3.6 Varying extension velocity

In a final variation on the reference model of narrow asymmetric rifting, we vary the total applied extension velocity and, therefore, plot source and host area against total extension instead of time (Figure 9d). Both source and host rock are more abundant when the extension velocity is lower. The lower the velocity, the slower the rift migration, the longer faults are active and the more marine and continental sediments can be deposited for a certain amount of extension. Thus, the thermal effects of greater burial depths and slower rift migration lead to source rock forming in most basins for an extension velocity of 5 mm yr$^{-1}$. Total sediment area is reduced for faster velocities, and the faster migrating rift also prohibits the formation of thick sediment packages. Therefore, no ore formation mechanisms occur for an extension velocity of 20 mm yr$^{-1}$ (Figure 9d bottom).







*Figure 8 (Previous page) Model evolution of wide rift simulation W-2 showing widespread deformation followed by rift migration and the transition to continental breakup. (a)-(f) Time series of zoom-ins focusing on FastScape topography (top) and sedimentary basins (bottom). The figures are always centred around x = 350 km. Black rectangles indicate the locations of further zoom-ins in (g)-(j). (g)-(j) Zoom-ins of specific basins with and without ore formation mechanisms. (i)-(j) Ore formation mechanism 3, where inactive faults connect source and host rock in the same basins. Note that the host rock in the left basin consists of deep marine sediments (silicates).*

## 4 Discussion

### 4.1 Predicted spatial and temporal mineralization windows

Our numerical models of continental rifting indicate that asymmetric narrow rifting is most likely to produce favourable conditions for CD-type metallogenesis. This rifting type creates the largest area of source rock and the most basins where ore formation mechanisms 1 and 2 can occur (Figure 4 and Figure 6). These scenarios occur in the major basin on the narrow margin, where the border fault is in turn active, inactive and reactivated until rift migration ceases. Fluids can either migrate along the border fault itself, or along smaller synthetic faults that develop in the major basin as the rift migrates. Prolonged activity of the border fault is important as it creates accommodation space for the deposition of a thick package of both continental and marine sedimentary rock, so that the lowermost units of that package reach the temperatures required to form a source rock as they are heated up by rising asthenosphere and are insulated by overlying sediments.

The distributed faulting and shorter-lived narrow faulting in the narrow symmetric rifting case does not allow for thick sedimentary sequences to form before oceanic spreading starts around 15 My, and only two small patches of source rocks form in one of the nine simulations (Figure 6&7). In five of the wide rifting simulations, a large basin forms once rift migration starts, but soon after source rocks begin to form, activity on its border fault ceases as the rift stabilizes. More common is ore forming mechanism 3, which occurs in the wide margin for eight out of the nine simulations (e.g., Figure 8e,f,i,j). Our simulations further show that the ore-forming potential depends on a favourable balance between rifting velocities, sedimentation rates and erosion efficiencies (Figure 9).

From the above, it is clear that the relative timing of border or synthetic fault activity, source rock generation and the deposition and burial of marine sediments determines the temporal window for mineralization. For example, in the narrow asymmetric simulation in Figure 3j&k and Figure 4a, marine sediments along the border fault have been heated up too much to be able to act as host rock by the time the fault is reactivated. Meanwhile the distal part of the major basin has received less sediment and rotated less along the border fault, such that the marine sediments stayed cool enough to act as host rock for the fluids potentially tapped by the new synthetic faults. Here, the temporal mineralization window for ore formation mechanism 1 is 3 My, and mineralization would occur in the upper several kilometres of sediment within 38-50 km from the shoreline. In the





example of ore formation mechanism 2 (Figure 4b), the border fault is reactivated just in time to connect source and host rock for ~3 My before all marine sediments along the fault have heated too much and the fault becomes inactive again. In this case, mineralization would occur closer to shore (~14 km) in the upper few kilometres of sediment. The example simulations for

mechanism 1 and 2 each also contain instances of the other mechanism (see Figure 4), with a temporal window of 2 and <1 My, respectively.

As discussed in Section 2.4, we assume that faults are more permeable than their surroundings and thus form upwards discharging fluid pathways. Recharge of fluids in the ore-producing basins is however also important; this can occur

downwards along faults. As hydraulic head can drive fluid flow and generally follows the water table that mimics the topography, recharge along the main border fault is likely (e.g., Ridley, 2013). However, along-strike, both upward and downward flow may occur along the same fault and therefore both ore formation mechanism 1, with multiple smaller faults cross-cutting the major basin, and mechanism 2, which relies on one larger fault are plausible mechanisms. Moreover, during both mechanisms other faults are also active in the same basin on the narrow margin (see Figure 4), providing additional

recharge pathways.

## 4.2 Comparison to observations and current mineral system concepts

Our physics-based simulations of coupled geodynamic and surface processes can be used to better understand the formation of CD-type deposits in well-studied provinces, which are often complicated by later tectonic overprints. The modelled ore formation mechanisms 1 and 2 reproduce some important characteristics of major Pb-Zn provinces. For one, in the Selwyn

Basin (Canada) and the Carpentaria Zinc Belt (Australia), up to several kilometres of sediments can vertically separate the source rocks from the host rocks (Blaikie and Kunzmann, 2020; Hayward et al., 2021; Rodríguez et al., 2021). This agrees well with the 1-3 km in between source and host rock at the time of mineralization (20 My) for ore formation mechanism 1 and 2 in Figure 4. Additionally, the source sequences themselves are 2-5 km thick in the McArthur (Australia) and Selwyn Basins (Cooke et al., 2000; Rodríguez et al., 2021 and references therein), as we find for mechanisms 1 and 2. Lastly, many

CD-type deposits are considered to have formed as the host unit was undergoing early to mid-burial diagenesis, i.e., at < 2 km depth (e.g., Hnatyshin et al., 2015; Hayward et al., 2021). The predicted depths of mineralization of 1-3.5 km in Figure 4 are of similar magnitude.



*Figure 9 Total source and host area over time for variations on the narrow asymmetric (NA) rift type, varying (a) marine sedimentation rate $R_M$, (b) marine diffusion coefficients $K_{M\ sand}$ and $K_{M\ silt}$, (c) river incision rate $K_f$, and (d) extension velocity $v_x$. Averages of each set of nine initial plastic strain configurations are shown per setup for easier comparison. Note the different ranges on the vertical axes and the total extension that is plotted instead of time in (d). Each graph is accompanied by a zoom-in of the rift basins for a simulation with the same initial strain configuration as the reference simulation in Figure 3, but with a higher value of the varied parameter instead of the reference value in bold. Increasing the marine sedimentation rate, the marine diffusion coefficients, and the river incision rate increases the total area of source rock at the end of a model run. Final host rock area increases for higher marine sedimentation rates and higher river incision rates, while it decreases for faster extension velocities. No ore formation mechanisms occur for the highest extension velocity.*





Fault reactivation or episodic activity along fault zones has been suggested for some districts (e.g., the Selwyn Basin, Lund, 2008). Some known Pb-Zn mineralization is identified as post-faulting, so deposition occurred after the associated fault had become inactive, but syn-rift, as the locus of deformation shifts throughout the rift's evolution (Walsh et al., 2018). In this case, faults still provide fluid pathways, either passively or through fault-valve behaviour driven by fluid overpressure (Walsh et al., 2018). These observations are mirrored by some of our results: The major border fault that acts as fluid pathway in ore

formation mechanism 2 (Figure 4) shows multiple episodes of inactivity and reactivation, which could provide a trigger for fault-valve behaviour by tectonic events. Furthermore, ore formation mechanism 3 in the wide margin would correspond to the post-faulting mineralization.

There are relatively few reliable temperature constraints for the fluids that formed CD-type deposits. In some localities,

homogenization temperatures of fluid inclusions trapped in vein assemblages from the fault-bound feeder zone record the minimum temperature of fluid entrapment of between 250 and 300 °C (Cooke et al., 2000; Rajabi et al., 2015; Magnall et al., 2016). These temperatures provide the lowest potential temperature constraints for fluids in the source rock. Within the temporal and spatial mineralization windows of mechanism 1 and 2 in Figure 4, the maximum source rock temperature is 508 and 455 °C, resp. These temperatures would provide enough heat to the fluids to enter the mineralization site at >250 °C,

where they would quickly cool due to mixing with cooler fluids.

Using Eq. (9), which includes a conversion of source rock area to volume, the potential endowment for the average of all nine narrow asymmetric simulations can be calculated (Figure 5d). As the majority of source rock in these simulations is concentrated in the major basin adjacent to the border fault in the narrow margin (see for example the source rock distribution

in Figure 3k), the total source rock area is representative of the source rock tapped by faults through mechanisms 1 and 2. These mechanisms were active between 13.5 to 21.0 My in Figure 4; this timeframe corresponds to a possible endowment of up to 69 Mt zinc and 16 Mt lead. This is of the same order of magnitude as the maximum currently estimated zinc tonnage of 27.4 Mt (Red Dog, Alaska) and the maximum lead tonnage of 28.0 Mt (Broken Hill, Australia) reported in the database of Hoggard et al. (2020).


Our results show that the co-occurrence of source rock, host rock and active faults occurs predominantly in the major basin in the narrow margin of asymmetric rift systems (ore formation mechanism 1 and 2). Possible sites of mineralization are located within several tens of kilometres of the shoreline at the basin margin (Figure 4), which supports a number of geological observations. For example, high salinity fluids derived from seawater evaporation along the basin margin are preserved in fluid

inclusions in some locations (Banks et al., 2002; Leach et al., 2004). In some of the Paleoproterozoic Australian deposits, mineralization is hosted by units that are transitional between shallow water, evaporitic facies and deep-water facies (Kunzmann et al., 2019). In terms of reduced sulfur, the metal trap in some of the largest and highest-grade deposits is derived primarily from sulfate reducing bacteria (e.g., Fallick et al., 2001), which are most active in regions of high carbon burial along





basin margins (Kallmeyer et al., 2012). Some of the largest deposits in the North American Cordillera are also hosted in
organic-rich units that were deposited in highly productive settings, similar to modern day upwelling zones (Dumoulin et al.,
2014). For example, trace fossil evidence from the Cordilleran Red Dog system also suggests that host rocks were deposited
on the middle to outer shelf (Reynolds et al., 2015). Notably, numerical modelling of mineralizing fluid flow in the Selwyn
Basin by Rodríguez et al. (2021) also shows larger areas of metal enrichment towards the higher topography and shallower
water of the carbonate platform. Here, cooling by recharging fluids and along the sloping source layer topography is most
efficient. In short, different lines of evidence point to the optimum alignment of ingredients for CD-type mineralization in the
near-shore marine domain.

Our numerical modelling results also indicate that rift migration is key for large CD-type systems. Migration generates a large,
deep, hot sedimentary basin on the narrow margin, in which episodic faulting connects source and host rocks leading to
favourable conditions for CD-type metallogenesis. Identifying narrow passive margins of ancient asymmetric rifts could
therefore help constrain the exploration space for CD-type deposits. For example, asymmetry of present-day conjugate margins
can be found in the Iberia-Newfoundland and the Central South Atlantic conjugates (Brune et al., 2017). The upper plate and
lower plate passive margin concept (Lister et al., 1986) has been used to describe the structural asymmetry of such conjugates.
In the Brazil/Angola-Gabon margins, the upper plate margin is narrow with a sharp crustal taper and little hyper-extension,
while lower plate margins exhibit hyper-extended and exhumed domains and a distal supra-detachment (Péron-Pinvidic et al.,
2017). Our model predicts that the potential for large endowments via ore formation mechanisms 1 and 2 is highest on the
upper plate side where the continental margin is narrow (i.e., the Brazilian Camamu basin and the Namibe/Benguela basins
offshore Africa), although there are additional secular factors (e.g., seawater chemistry) that might limit margin fertility in
modern settings (Wilkinson, 2014). There are not many ancient margins where upper or lower plate sections have been
identified, reflecting a need for additional geophysical data to characterize ancient margin structure. In the ancient western
Laurentian margin (North America), however, both plate types have been recognised, and both types host CD-type deposits
(Lund, 2008; Hayward and Paradis, 2021). This occurrence of mineralization on both margin types could reflect a combination
of all three ore formation mechanisms occurring, with mechanism 3 dominant on the lower plate side.

## 5 Conclusions

This study is the first to propose a framework for the first-order identification of favourable conditions for sediment-hosted
ore-formation systems using coupled numerical models of rift dynamics and surface processes. Using this framework, we
identified three mechanisms that combine the main components of the CD-type mineral system (metal source, fluid conduit
and metal trap/host) in continental rifts: 1) smaller, younger faults deform a sedimentary basin and connect the potential source
and host rock that have formed in the active basin; 2) an active fault provides a pathway from source to host rock that formed

in the basin created by this fault; and 3) source and host rock are generated in a basin that is no longer active, but whose inactive fault could still act as a fluid conduit either through free fluid convection or during fault reactivation.

The time during and space in which these three ore formation mechanisms are active – the temporal and spatial mineralization window – is small. Mineralization windows occur most often in the narrow continental margin of narrow asymmetric rift
systems, where over a time period of less than 3 My active faults provide a pathway for fluid to migrate from potential source rocks to potential host rocks. The predicted mineralization then occurs within several tens of kilometres from the shore line within the top several kilometres of sediment. Temporal constraints on the mineralization window arise from conflicting results of heating and fault activity. For example, the heating of basal coarse continental sediments by rising asthenosphere and continuous deposition of new sediments that blanket the units below generates source rocks, but simultaneously heats up
shallower marine sediments too much to act as host rocks.

Rift migration, especially during narrow asymmetric rifting, promotes the formation of one deep basin along the persistent border fault on the narrow margin. Its thick sedimentary infill is continuously heated from below and generates large amounts of source rock that could produce metal endowment of similar magnitudes as the world's largest deposits. Moreover, all three
ore formation mechanisms occur in the narrow margin during rift migration, with the border fault acting as fluid conduit in mechanism 2 and 3 and the sequential faults of the migrating rift centre providing fluid pathways according to mechanism 1. The spatiotemporal window of mineralization during rift migration is modulated by the rate of sediment erosion and deposition as well as the extension velocity, stressing the importance of coupled geodynamics and surface processes for a holistic understanding of sediment-hosted mineral systems. The amount of source rock produced in narrow symmetric rifts and wide
rifts is only one-third of that generated in narrow asymmetric rifts, and favourable conditions for ore formation occur less often. Therefore, exploration for CD-type deposits could benefit from the identification of ancient narrow margins.

**Code availability**

The versions of ASPECT and FastScape used in this paper can be found in Zenodo repository 10.5281/zenodo.10048075 (Glerum, 2023). This includes the modifications made to ASPECT 2.4-pre commit
84d40e745328f62df1a09e15a9f1bb4fdc86141a (Bangerth et al., 2021b, a) and FastScape commit 18f25888b16bf4cf23b00e79840bebed8b72d303 (Bovy, 2020).

**Data availability**

Input files and output text files for each of the simulations presented are available from Zenodo repository 10.5281/zenodo.10048075 (Glerum, 2023), as well as raw visualization files for each of the snapshots of the simulations shown



in the figures. The repository also includes the python and ParaView postprocessing scripts used to create the graphs and
individual model snapshots.

**Author contribution**

All authors formulated the ideas and research goals for this manuscript. AG and SB designed the numerical experiments and
AG extended the model code and performed the simulations. Analysis of the model data was done by AG with input from JM.
AG visualized the model results and prepared the manuscript with contributions from all co-authors.

**Competing interests**

The authors declare that they have no conflict of interest.

**Acknowledgements**

We thank the Computational Infrastructure for Geodynamics (geodynamics.org), which is funded by the National Science
Foundation under award EAR-0949446 and EAR-1550901, for supporting the development of ASPECT. We also gratefully
acknowledge the computing time granted by the Resource Allocation Board and provided on the supercomputer Lise at
NHR@ZIB as part of the NHR infrastructure. The calculations for this research were conducted with computing resources
under the project bbp00039. AG and SG are funded by a Helmholtz Recruitment Initiative. Figures were created using
Scientific colour maps 7 (Crameri, 2021; Crameri et al., 2020), ParaView (Ahrens et al., 2005; Ayachit, 2015), MATLAB
(The MathWorks Inc., 2022), python (The Python Software Foundation, 2023) and Inkscape (Inkscape Team, 2022).

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
