# Peer review of "Geodynamic controls on clastic-dominated base metal deposits"

_EGUsphere, 2023_

## Author Comment (AC1)

**Response to the reviewers**

We thank both reviewers for their positive and constructive remarks. Below we address each of the points of reviewer 1 and show how we adapted the manuscript accordingly. These and other small changes to the manuscript are also indicated in red in the updated version of the manuscript.

Reviewer 1 - Mark Hoggard

Glerum et al. have coupled rift-modelling software developed in ASPECT to a surface-processes model called FastScape. They use this computational infrastructure to perform a suite of 2D numerical models of extensional-basin formation and sedimentation under different rifting scenarios. They apply a mineral-systems approach to identify characteristics that might be favourable to the formation of clastic-dominated lead-zinc deposits. Their results indicate that architectures involving narrow, asymmetric rifts are most likely to produce a thick pile of sediments that (a) contain suitable lithologies necessary to act as both source and host rocks; (b) have the correct thermal histories; and (c) can be connected to one-another by suitable fluid pathways along faults. They conclude that identification of ancient, narrow margins in the geological record will likely help to reduce the search-space for new deposit discoveries.

I enjoyed reading this thought-provoking manuscript and am impressed by the amount of work that it contains. It certainly raises new possibilities concerning which mechanisms may be responsible for the observed localisation of these deposits within very specific regions. I am in support of publication as is, but have a few optional suggestions and talking points that may potentially add to the story. Feel free to take or leave as you see fit and thanks for the opportunity to read it!

**A few general thoughts:**

My interpretation of the numbers in Lines 377-378 and focus of Figure 5 is that it is mainly the generation of appropriate "source rocks" and the pathways that connect them to host rocks that are the limiting factor across the different types of rift (i.e. the availability of suitable host rocks is not generally a problem). Is this the case? I don't think that point is made particularly clearly anywhere (my apologies if this is incorrect). You could potentially also include a host-rock-only count in Figure 6?

Host rocks are indeed more widespread than source rock in all simulations (compare host rock over time plots below in Fig. R1 with the plots in Fig. 5). That said, the majority of host rocks are found in the wide continental margins without active faults that formed by rift migration in narrow asymmetric and wide rifts, and there are basins with source rock without host rocks. The latter mostly occurs 1) when marine sediments which were previously cold enough to act as host rocks are heated up upon burial in the border basin, which closes the mineralization window, 2) after rift migration has stopped and, thus, faults in the border basin have become inactive, which also closes the mineralization window, or 3) during active faulting in the rift

centre. These cases are exemplified by the smaller amount of orange than red bars in Fig. 6 (although these bars do not necessarily represent the same timestep). Therefore, suitable host rock does act as a limiting factor. The focus of Fig. 5 lies on source rock area as it is less widespread and determines the size of the potential deposit, and the size of the host rock area is less important than its presence, as you also recognize in your next point.

We think adding the count of basins with host rock only will make it harder to get the message of Fig. 6 across, as it will require increasing the upper limit of the y-axis. We have however added the following statement to the main text (starting line 422):

*Note that there are also basins with host rocks and active faults but no source rocks (e.g., Figure 5a at 15 My) and basins with only host rocks and inactive faults (e.g., border basin in Figure 3i and in all wide margins). These cases are however not represented in Figure 7, as they do not have a potential metal source and can therefore not be a site of mineralization.*

[Figure]

*Figure R1 Host rock area over time for narrow asymmetric, narrow symmetric and wide rift simulations. Grey shading indicates the start of oceanic spreading.*

Related to this point, do we actually want widespread host rocks? Local ones that are in the vicinity of the brine conduits are probably all we need to concentrate metals into a viable deposit. Having them widespread has the potential to result in metals precipitating in lower concentrations over larger areas, depending on the particular hydrothermal circulation pathways.

The host rocks provide a key ingredient for ore formation, i.e., an abundant source of reduced sulfur. We agree that *widespread* host rocks are not required. However, in the geological record it is clear that mudstones are ubiquitous in most basins worldwide, and yet large Zn deposits are rare. This points to the importance of a focussing mechanism of the fluids towards the host rock. We regard fluid focusing through active or inactive fault structures as one of the key parameters in our analysis, because it may help avoid widespread disseminated mineralization at low grades within the abundant host rock units.

We would also like to note that with increased marine sedimentation rate, not only does the modelled amount of host rock become more widespread, but the amount of source rock also increases (see Section 3.4.1), in turn increasing the potential for metallogenesis.

We have added the following to the Methods section 2.4 (line 206) to highlight the focussing effect of faults:

*Fault activity (e.g., seismic pumping and pulsing and fault valving) and passive upward flow then bring the fluids to the surface (Walsh et al., 2018), **mechanisms that focus the fluids and concentrate the metals in the host rock.***

In the Abstract (line 23) and PLS (line 31-32), we added:

*… and 4) the generation of smaller faults that cut the major basin created by the border fault and provide additional pathways **for focussed fluid flow** from source to host rock.*

*We use computer simulations of rifting and the associated erosion and deposition of sediments to understand why they formed in some basins, but not in others. **In particular, we look for basins that combine metal source rocks, faults that focus fluid flow and rocks that can host deposits.***

I realise that these models do not include melting and it is therefore prudent to not overly speculate on this aspect, but I wonder if they might also provide insights on the potential for generating mafic volcanism? These rocks often act as a source of Zn and particularly Cu, such as the Eastern Creek Volcanics at Mount Isa. Given that the locations and strength of mantle upwelling is modelled, we can infer where in the model domain the solidus is most likely to be crossed. It strikes me that the narrow asymmetric margin more often has a focused mantle upwelling in its vicinity than other margins in the various models? They may therefore be more susceptible to generating volcanic rocks, which would be another potential strength of this particular geodynamic setting.

Indeed, our simulations do not include melting and represent magma-poor rifts. As you suggested, we computed the degree of melting in a narrow asymmetric rift simulation based on a linear interpolation of the melt fraction between a solidus and liquidus of Katz et al. (2003). As shown in Fig. R2 at 20 My, the melt fraction is highest in the asthenosphere underneath the narrow asymmetric margin during rift migration. If we vertically integrate the melt fraction, we estimate melts could generate a magmatic layer of up to 1.5 km thickness (e.g., Brune et al., 2014). Only some part of this melt would intrude the basement or sedimentary sequence.

[Figure]

*Figure R2 Melt fraction at the time of ore formation mechanism 1 (20 My) for whole model domain (bottom) and zoom-in (top) for narrow-asymmetric rifting simulation NA-4.*

A further potential benefit of the narrow-margin setting would be proximity of source and host rocks to the adjacent continental platform. This architecture would seem to be ideal for the "brine factory" ideas of Manning & Emsbo (2018) and others – i.e. potential for a nearby evaporative platform to generate brines and a hydrological head to drive them down into the basin.

Thank you for this suggestion. We have added the brine reflux flow system hypothesis of Manning and Emsbo (2018) to the Discussion where we discuss other evidence for metallogenesis close to shore in Section 4.2 (line 588):

*For example, high salinity fluids derived from seawater evaporation along the basin margin are preserved in fluid inclusions in some locations (Banks et al., 2002; Leach et al., 2004),* ***inspiring the hypothesis of a platform "brine factory" (e.g., Leach et al., 2010).***

…

Line 598 onwards:

***Modelling of the brine reflux system by Manning and Emsbo (2018) shows that evaporated seawater generated on a carbonate platform (the brine factory) can form ore fluids of the required salinity and temperature that sink into the platform, flow oceanward and discharge along faults to the nearby ocean floor.***

Please forgive the personal bias, but one of the interpretations we made in Hoggard et al. (2020) was that large clastic-dominated base metal deposits are all located in basins that occur on cratonic margins. The models in this study are all run using regular, Phanerozoic-style (ie. thin) lithosphere as the starting template. It may be that you don't agree with the craton story, but I'd be interested to hear your take on

which aspects of the narrow-margin fertility might translate well to a potential extensional setting on the edge of thick lithosphere? In addition, which aspects of your model results might change if they were run using thicker, potentially depleted lithosphere? For example, I would guess that the basal heat flow would be lower and it would be trickier to get the source rocks into your desired temperature window? At the same time, the solubility of Zn and Pb in brines is still high at lower temperatures, as long as they are oxidised, so I don't think it would matter much.

We were actually very intrigued by your 2020 study. Our approach has been to investigate the controls on mineralization with models of continental rifting of normal continental lithosphere and then to run a second suite of simulations in the presence of a craton, to fully explore the possible craton effects. The latter results warrant a separate publication, but we can say that a craton very close to the initial rift centre determines the direction of rift migration (conform Raghuram et al., 2023) and, therefore, affects the potential of the border basins for metallogenesis.

We have added the following sentence to the Discussion (section 4.2) to point out this important direction of future research (line 579):

*Future modelling should also investigate the role of thick, cold cratonic lithosphere (e.g., Raghuram et al., 2023; Gouiza and Naliboff, 2021) in promoting the formation and/or preservation of CD-type deposits, as Hoggard et al. (2020) (and Groves and Santosh, 2021; Lawley et al., 2022; Huston et al., 2022) found a correlation between deposit location and the edges of thick and/or cratonic lithosphere.*

**Some specific comments:**

The Plain Language Summary makes mention of the 3 Myr maximum extent for mineralisation, but this fact is not listed in the abstract. It seems potentially important enough to mention somewhere in there too?

Good point. We have added it to the abstract as follows (line 15):

*We show that the largest potential metal endowment can be expected in narrow asymmetric rifts,* **where the mineralization window spans about 1-3 My in the upper ~4 km of the sedimentary infill close to shore**.

Mccuaig should be McCuaig throughout.

Thank you, we have fixed the capitalization throughout.

Just checking - is the sand and silt porosity set to zero, as suggested in Table 1? I only ask because I think that renders the e-folding depths redundant.

That is correct, the sand and silt porosity are set to zero and therefore the e-folding depth has no effect. I have removed the mention of e-folding depth from line 183 of section 2.3 and from Table 1.

Lines 294-295: Qualify that the "geotherm and adiabat *temperatures* match".

We have changed the line to

*… such that at the LAB the geotherm and adiabat **temperatures** match **(at 1350 °C)***.

Line 326: Typo "inhered" → "inherited"?

Thank you for spotting this typo.

Figs 3, 7 and 8: Add a simple label above panels stating the model type (e.g. "Narrow Asymmetric Rift") to ease quick comparison.

We have labelled the figures with their respective rift type.

Figure 4: I found the panels for ore formation mechanisms 2 and 3 a bit small – the first one is better. I also wonder if you could potentially reverse the domain for model NA-9 to make comparison and contrast to NA-4 easier (i.e. have all the rifts migrating in the same direction)?

We agree. I have split Figure 4 into two separate figures, one for mechanism 1 and one for mechanism 2 and 3. I have also mirrored the figures for model NA-9 so that the narrow margin also occurs on the right-hand-side of the model domain.

Figure and text order: Potentially move Figures 7 and 8 ahead of 5 and 6, so a reader sees all the model results prior to the scenario comparison figures. Lines 410-415, Figures 5 and 6 would then naturally fall under a separate subsection after 3.3 called something like "Optimal characteristics for mineralisation" or similar.

Thank you for the suggestion; we have however refrained from reordering the figures and text to keep the flow of the text and to pique the reader's interest earlier on.

There currently is no Section 3.4.

Thank you, we have corrected the numbering.

Line 455: typo "at in".

We have fixed it to "in".

**References**

Brune, S., Heine, C., Pérez-Gussinyé, M., and Sobolev, S. V.: Rift migration explains continental margin asymmetry and crustal hyper-extension, Nature Communications, 5, 4014, https://doi.org/10.1038/ncomms5014, 2014.

Gouiza, M. and Naliboff, J. B.: Rheological inheritance controls the formation of segmented rifted margins in cratonic lithosphere, Nature Communications, 12, 1–9, https://doi.org/10.1038/s41467-021-24945-5, 2021.

Groves, D. I. and Santosh, M.: Craton and thick lithosphere margins: The sites of giant mineral deposits and mineral provinces, Gondwana Research, 100, 195–222, https://doi.org/10.1016/j.gr.2020.06.008, 2021.

Hoggard, M. J., Czarnota, K., Richards, F. D., Huston, D. L., Jaques, A. L., and Ghelichkhan, S.: Global distribution of sediment-hosted metals controlled by craton edge stability, Nature Geoscience, 13, 504–510, https://doi.org/10.1038/s41561-020-0593-2, 2020.

Huston, D. L., Champion, D. C., Czarnota, K., Duan, J., Hutchens, M., Paradis, S., Hoggard, M., Ware, B., Gibson, G. M., Doublier, M. P., Kelley, K., McCafferty, A., Hayward, N., Richards, F., Tessalina, S., and Carr, G.: Zinc on the edge—isotopic and geophysical evidence that cratonic edges control world-class shale-hosted zinc-lead deposits, Miner Deposita, https://doi.org/10.1007/s00126-022-01153-9, 2022.

Katz, R. F., Spiegelman, M., and Langmuir, C., H.: A new parameterization of hydrous mantle melting, Geochemistry Geophysics Geosystems, 4, 1–19, https://doi.org/10.1029/2002GC000433, 2003.

Lawley, C. J. M., McCafferty, A. E., Graham, G. E., Huston, D. L., Kelley, K. D., Czarnota, K., Paradis, S., Peter, J. M., Hayward, N., Barlow, M., Emsbo, P., Coyan, J., San Juan, C. A., and Gadd, M. G.: Data–driven prospectivity modelling of sediment–hosted Zn–Pb mineral systems and their critical raw materials, Ore Geology Reviews, 141, 104635, https://doi.org/10.1016/j.oregeorev.2021.104635, 2022.

Raghuram, G., Pérez-Gussinyé, M., Andrés-Martínez, M., García-Pintado, J., Araujo, M. N., and Morgan, J. P.: Asymmetry and evolution of craton-influenced rifted margins, Geology, 51, 1077–1082, https://doi.org/10.1130/G51370.1, 2023.

Walsh, J. J., Torremans, K., Güven, J., Kyne, R., Conneally, J., and Bonson, C.: Fault-Controlled Fluid Flow Within Extensional Basins and Its Implications for Sedimentary Rock-Hosted Mineral Deposits, in: Metals, Minerals, and Society, 237–269, 2018.

---

## Author Comment (AC2)

**Response to the reviewers**

We thank both reviewers for their positive and constructive remarks. Below we address each of the points of reviewer 2 and show how we adapted the manuscript accordingly. These and other small changes to the manuscript are also indicated in red in the updated version of the manuscript.

Reviewer 2 - Tim Jones

Overall that paper provides valuable insights into the dynamics of clastic-dominated base metal deposits. The authors cover a lot of ground to bring together geodynamics and a mineral systems analysis, providing an approach itself that will be valuable reading to multiple disciplines.

The modeling results provide some testable conjectures, such as "[the] conditions for deposit formation can briefly occur in both narrow and wide rifts for at most 3 My", and that narrow, asymmetric rifts are the most fertile settings for these deposits. The paper contains a thorough discussion of the results that includes comparisons to real geological cases. It lists some assumptions of the model but lacks a discussion of how those assumptions may limit the specific model results and applicability of the conclusions drawn. Great value from the models could be gained by the reader if the temperature field was included in the result images.

Some minor comments to address below.

"In sedimentary basins, key components of the mineral systems model are the metal source, flow conduits for metal-bearing fluids, and a trap for concentrating metals at the deposit site."
It would be helpful to see some references here, and explicitly state that if these are the 'key' processes, what are the secondary processes deemed to be less important to focus on and why.

We have added two references, the first describes the genetic model, the second the mineral system components of CD-type deposits (line 54-55):

*In sedimentary basins, key components of the mineral systems model are the metal source, flow conduits for metal-bearing fluids, and a trap for concentrating metals at the deposit site* **(e.g., Wilkinson, 2014; Lawley et al., 2022)**.

Lawley et al. (2022) for example also list preservation and direct detection as components of the CD-type deposit mineral system. These components are not key to the formation, but to the survival and discovery of the deposits, and are not considered in this paper. We described the key components mentioned – metal source, fluids, conduits and trap – in more detail in the paragraph that follows the quoted sentence. That said, every deposit experiences site-specific circumstances

that can be considered secondary components and cannot be addressed by the general nature of our simulations.

In the Discussion (line 579), we now point to one future avenue to include one of these secondary components, the preservation of deposits along cratonic edges:

*Future modelling should also investigate the role of thick, cold cratonic lithosphere (e.g. Raghuram et al., 2023; Gouiza and Naliboff, 2021) in promoting the formation and/or preservation of CD-type deposits, as Hoggard et al. (2020) (and Groves and Santosh, 2021; Lawley et al., 2022; Huston et al., 2022) found a correlation between deposit location and the edges of thick and/or cratonic lithosphere.*

"Eq (9) $dz$ = 20 km is the assumed in-plane extent of the basins to create a source rock volume"
Assumed based on what? This is somewhat important when you are comparing model results to estimates of deposit endowment to validate your results since I don't see why this value couldn't be arbitrarily set much higher or lower. Unless it is calibrated using realistic volume estimates? In which case it can't be used to justify the resulting endowments from the models.

We did not calibrate the assumed in-plane extent to known endowment, and agree that the arbitrary value of 20 km could be smaller. For example, the complex subbasin scale in the southern McArthur Basin (Fig. 11d of Blaikie and Kunzmann 2020), varies from ~1.5 to 15 km. Manning and Emsbo (2018) assumed a fault trace length of 10 km to estimate the total mass of Zn and Pb in their simulations of the brine reflux system. For comparability, we have therefore decided to adopt the value of 10 km and refer to the above papers in the text. However, note that we have to make assumptions about the other parameters that feed into the computation of the endowment (Eq. 9) as well. For example, we assume that 65% of the metals is leached from the source rock, as described in the Methods section 2.4.

We have adapted the following sentence in the Discussion to emphasize the assumptions we make in the endowment calculation (line 571):

*Using Eq. (9), which includes a conversion of source rock area to volume **assuming an in-plane extent of the basins of 10 km**, the potential endowment for the average of all nine narrow asymmetric simulations can be **estimated** (Figure 6d).*
*…*
*These mechanisms were active between 13.5 to 21.0 My in Figure 4; this time frame corresponds to a possible endowment of up to **35** Mt zinc and **8** Mt lead. This is of the same order of magnitude as the maximum currently estimated zinc tonnage of 27.4 Mt (Red Dog, Alaska) and **similar to** the maximum lead tonnage of 28.0 Mt (Broken Hill, Australia) reported in the database of Hoggard et al. (2020). **Endowment estimates would improve from resolving the relevant processes in 3D and from the inclusion of fluid flow.***

We have also replotted the endowment figure for an in-plane extent of 10 km, and added the above cited references to the text in Section 2.4 as follows (line 253):

*where ρ is the rock density, $dz = 10km$ is the assumed in-plane extent of the basins to create a source rock volume* **(based on basin extent in Blaikie and Kunzmann (2020) and on the assumptions of Manning and Emsbo (2018))***,*

One limitation not discussed is the modelling of near surface deformation, which is a brittle process, using equations that treat the Earth as a viscous fluid. I understand that this is common practice but strictly speaking it should not apply to the upper crust, and is even debatable in some situations at greater depths. Since a portion of the results depend on the model's ability to simulate near-surface faulting, I suggest adding a discussion around this to the limitations section.

Indeed we use a highly-viscous fluid description of the Earth, which is warranted by the large timescales we are interested in compared to the Maxwell relaxation time. We modify the viscosity in the governing equations into an effective viscosity that takes into account plasticity (approximation of brittle faulting). As the reviewer states, this is common practice. Comparison of these viscous-fluid approximation methods to analog models, analytical solutions of failure and to geomechanic codes shows good agreement, also for ASPECT (Buiter et al., 2006; Kaus, 2010; Buiter et al., 2016; Glerum et al., 2018; Duretz et al., 2019). A large body of thermomechanical modelling work uses this approach to look at upper crust deformation, and we have therefore limited the discussion of this common approach to the following sentence in Section 2.5 Model assumptions (line 283):

*For one***, the governing equations approximate the Earth as a highly-viscous fluid on geological timescales, modifying the viscosity to allow for non-viscous behaviour such as brittle deformation. Second***, our simulations are 2D*

Is there a missing section 3.4 or just a typo?

This was a typo, thank you, we have corrected the numbering of the sections.

Would be great to see thermal evolution here in the results, alongside composition and strain. Can you say why you thought it was not important to include that? It would help provide some insight into where melting might be focused even if not explicitly predicted.

As can be seen in the figures, we did include two crustal isotherms, those of 150 and 250 °C, as these delineate the upper boundary of the host rock temperature window and the lower boundary of the source rock temperature window, respectively. Before submission we tested the inclusion of more crustal isotherms, but this made the figures very crowded. As the majority of CD-type deposits have sedimentary rocks as metal source instead of volcanics, and we do not include melting and melt transport, we did not focus on possible melt regions.

That said, we have now reprocessed figures 3a-f to include isotherms in the lithosphere and asthenosphere. We also postprocessed one narrow asymmetric rift simulation to compute the melt fraction based on a linear interpolation between a solidus and liquidus of Katz (2003), see Figure R1 and the response to reviewer 1.

Potential melting would be focused underneath the oceanward side of the narrow margin, i.e., in the right location for intrusion into the basin where the most favourable ore formation mechanisms occur.

[Figure]

*Figure R1 Melt fraction at the time of ore formation mechanism 1 (20 My) for whole model domain (bottom) and zoom-in (top) for narrow-asymmetric rifting simulation NA-4.*

The sedimentation rates and volumes are discussed and compared with observations but not the resulting sedimentary structure of the basins. This seems like a key prediction of the models also. Do you have any insight into how, in a broad sense, does the predicted sequence of lithologies and their distribution compare to observations?

A direct comparison of our model lithologies and observed lithostratigraphies of known large deposits is hindered by the limited number of sediment types in the simulations and the difference in resolution. For example, a description of the Barney Creek Formation (host to the Teena and McArthur River deposits) in Hayward et al. (2021) has a stratigraphic resolution of several 10s of meters, while the maximum resolution of our simulations is 313 m. Our simulations resolve three types of sediments, predominantly coarse continental (sandstone), predominantly fine continental (siltstone), and predominantly marine sediments. The stratigraphy of the Umbolooga Subgroup, to which the Barney Creek Formation belongs, is drawn by Blaikie and Kunzmann (2020, Fig. 2) as an alternating sequence of siliciclastics, mixed siliciclastics/carbonates, and carbonates. This agrees with the sequence of predominantly sandstone and marine sediments we predict in the fertile basins.

We have added the following statement to the manuscript (line 546):

***In the McArthur Basin, Blaikie and Kunzmann (2020) describe the Umbolooga Subgroup as an alternating stratigraphic sequence of siliciclastics, mixed siliciclastics/carbonates, and carbonates, which agrees with the level of stratigraphic detail we can resolve in our simulations.***

**References**

Blaikie, T. N. and Kunzmann, M.: Geophysical interpretation and tectonic synthesis of the Proterozoic southern McArthur Basin, northern Australia, Precambrian Research, 343, 105728, https://doi.org/10.1016/j.precamres.2020.105728, 2020.

Buiter, S. J. H., Babeyko, A. Y., Ellis, S., Gerya, T. V., Kaus, B. J. P., Kellner, A., Schreurs, G., and Yamada, Y.: The numerical sandbox: comparison of model results for a shortening and an extension experiment, in: Analogue and Numerical Modelling of Crustal-Scale Processes, vol. 253, edited by: Buiter, S. J. H. and Schreurs, G., Geological Society, London, Special Publications, 29–64, 2006.

Buiter, S. J. H., Schreurs, G., Albertz, M., Gerya, T. V., Kaus, B., Landry, W., le Pourhiet, L., Mishin, Y., Egholm, D. L., Cooke, M., Maillot, B., Thieulot, C., Crook, T., May, D., Souloumiac, P., and Beaumont, C.: Benchmarking numerical models of brittle thrust wedges, Journal of Structural Geology, 92, 140–177, https://doi.org/10.1016/j.jsg.2016.03.003, 2016.

Duretz, T., de Borst, R., and Le Pourhiet, L.: Finite Thickness of Shear Bands in Frictional Viscoplasticity and Implications for Lithosphere Dynamics, Geochemistry, Geophysics, Geosystems, 20, 5598–5616, https://doi.org/10.1029/2019GC008531, 2019.

Glerum, A., Thieulot, C., Fraters, M., Blom, C., and Spakman, W.: Nonlinear viscoplasticity in ASPECT: Benchmarking and applications to subduction, Solid Earth, 9, 267–294, https://doi.org/10.5194/se-9-267-2018, 2018.

Groves, D. I. and Santosh, M.: Craton and thick lithosphere margins: The sites of giant mineral deposits and mineral provinces, Gondwana Research, 100, 195–222, https://doi.org/10.1016/j.gr.2020.06.008, 2021.

Hayward, N., Magnall, J. M., Taylor, M., King, R., McMillan, N., and Gleeson, S. A.: The Teena Zn-Pb Deposit (McArthur Basin, Australia). Part I: Syndiagenetic Base Metal Sulfide Mineralization Related to Dynamic Subbasin Evolution, Economic Geology, 116, 1743–1768, https://doi.org/10.5382/econgeo.4846, 2021.

Hoggard, M. J., Czarnota, K., Richards, F. D., Huston, D. L., Jaques, A. L., and Ghelichkhan, S.: Global distribution of sediment-hosted metals controlled by craton edge stability, Nature Geoscience, 13, 504–510, https://doi.org/10.1038/s41561-020-0593-2, 2020.

Huston, D. L., Champion, D. C., Czarnota, K., Duan, J., Hutchens, M., Paradis, S., Hoggard, M., Ware, B., Gibson, G. M., Doublier, M. P., Kelley, K., McCafferty, A., Hayward, N., Richards, F., Tessalina, S., and Carr, G.: Zinc on the edge—isotopic and geophysical evidence that cratonic edges control world-class shale-hosted zinc-lead deposits, Miner Deposita, https://doi.org/10.1007/s00126-022-01153-9, 2022.

Katz, R. F., Spiegelman, M., and Langmuir, C., H.: A new parameterization of hydrous mantle melting, Geochemistry Geophysics Geosystems, 4, 1–19, https://doi.org/10.1029/2002GC000433, 2003.

Kaus, B.: Factors that control the angle of shear bands in geodynamic numerical models of brittle deformation, Tectonophysics, 484, 36–47, 2010.

Lawley, C. J. M., McCafferty, A. E., Graham, G. E., Huston, D. L., Kelley, K. D., Czarnota, K., Paradis, S., Peter, J. M., Hayward, N., Barlow, M., Emsbo, P., Coyan, J., San Juan, C. A., and Gadd, M. G.: Data–driven prospectivity modelling of sediment–hosted Zn–Pb mineral systems and their critical raw materials, Ore Geology Reviews, 141, 104635, https://doi.org/10.1016/j.oregeorev.2021.104635, 2022.

Manning, A. H. and Emsbo, P.: Testing the potential role of brine reflux in the formation of sedimentary exhalative (sedex) ore deposits, Ore Geology Reviews, 102, 862–874, https://doi.org/10.1016/j.oregeorev.2018.10.003, 2018.